# Late Oligocene astronomically paced contourite sedimentation in the Wilkes Land margin of East Antarctica: insights into paleoceanographic and ice sheet configurations

Keywords:

**Late Oligocene**

**Paleoceanography**

**Antarctic Ice sheet**

**Contourites**

**Obliquity**

Ariadna Salabarnada[1], Carlota Escutia[1], Ursula Röhl[2], C. Hans Nelson[1], Robert McKay[3], Francisco J. Jiménez-Espejo[4], Peter K. Bijl[5], Julian D. Hartman[5], Stephanie L. Strother[6], Ulrich Salzmann[6], Dimitris Evangelinos[1], Adrián López-Quirós[1], José Abel Flores[7], Francesca Sangiorgi[5], Minoru Ikehara[8], Henk Brinkhuis[5, 9]

[1]Instituto Andaluz de Ciencias de la Tierra, CSIC-Univ. de Granada, Armilla, 18100, Spain

[2]MARUM - Center for Marine Environmental Sciences, University of Bremen, Leobener Strasse 8, 28359 Bremen, Germany

[3]Antarctic Research Centre, Victoria University of Wellington, Wellington, 6140, New Zealand

[4]Department of Biogeochemistry, Japan Agency for Marine-Earth Science and Technology, Yokosuka, Kanagawa, 237-0061, Japan

[5] Department of Earth Sciences, Marine Palynology and Palaeoceanography, Faculty of Geosciences, Laboratory of Palaeobotany and Palynology, Utrecht University, Princetonlaan 8a, 3584 CB Utrecht, The Netherlands

[6]Department of Geography and Environmental Sciences, Faculty of Engineering and Environment, Northumbria University, Newcastle upon Tyne NE1 8ST, UK

[7]Department of Geology, University of Salamanca, Salamanca, 37008, Spain

[8]Center for Advanced Marine Core research, Kochi University, Nankoku, Kochi, 783-8502, Japan

[9]NIOZ, Royal Netherlands Institute for Sea Research, and Utrecht University, Landsdiep 4, 1797SZ 't Horntje, Texel, The Netherlands

*Correspondence to*: Ariadna Salabarnada (a.salabarnada@csic.es)

**Abstract**

**Antarctic ice sheet and Southern Ocean paleoceanographic configurations during the late Oligocene are not well resolved. They are however important to understand the influence of high-latitude Southern Hemisphere feedbacks on global climate under $CO_2$ scenarios (between 400 and 750 ppm) projected by the IPCC for this century, assuming unabated $CO_2$ emissions. Sediments recovered by the Integrated Ocean Drilling Program (IODP) at Site U1356, offshore of the Wilkes Land margin in East Antarctica, provide an opportunity to study ice sheet and paleoceanographic configurations during the late Oligocene (26-25 Ma). Our study, based on a combination of sediment facies analysis, magnetic susceptibility, density, and X-Ray Fluorescence geochemical data, shows that glacial and interglacial sediments are continuously reworked by bottom-currents, with maximum velocities occurring during the interglacial periods. Glacial sediments record poorly ventilated, low-oxygenation bottom water conditions, interpreted to result from a northward shift of westerly winds and surface oceanic fronts. Interglacial sediments record more oxygenated and ventilated bottom water conditions and strong current velocities, which suggests enhanced mixing of the water masses as a result of a southward shift of the Polar Front. Intervals with preserved carbonated nannofossils within some of the interglacial facies are interpreted to form under warmer paleoclimatic conditions when less corrosive warmer northern component water (e.g. North Atlantic sourced deep water) had a greater influence on the Site. Spectral analysis on the late Oligocene sediment interval show that the glacial-interglacial cyclicity and related displacements of the Southern Ocean frontal systems between 26-25 Ma were forced mainly by obliquity. The paucity of iceberg rafted debris (IRD) throughout the studied interval contrasts with earlier Oligocene and post-Miocene Climate Optimum sections from Site U1356 and with late Oligocene strata from the Ross Sea, which contain IRD and evidence for coastal glaciers and sea ice. These observations, supported by elevated sea surface paleotemperatures, the absence of sea-ice, and reconstructions of fossil pollen between 26 and 25 Ma at Site U1356, suggest that open ocean water conditions prevailed. Combined, these evidences suggest that glaciers or ice caps likely occupied the topographic highs and lowlands of the now marine Wilkes Subglacial Basin (WSB). Unlike today, the continental shelf was not over-deepened and thus ice sheets in the WSB were likely land-based and marine-based ice sheet expansion was likely limited to coastal regions.**

## 1. Introduction

Today, ice sheets on Antarctica contain about 26.5 million cubic kilometres of ice, which has the potential for raising global average sea level by 58 m, with the East Antarctic Ice Sheet constituting 53.3 m of this sea level equivalent (Fretwell et al., 2013). Satellite observations indicate significant rates of change in most of the West Antarctic Ice Sheet (WAIS) and some sectors of the East Antarctic Ice Sheet (EAIS). These include thinning at their seaward margins (Pritchard et al., 2012) and accelerating ice shelves basal melt rates (Rignot et al., 2013; Shen et al., 2018). Given the uncertainties in projections of future ice sheet melt, there has been a growing number of studies of sedimentary sections from the surrounding margins of Antarctica targeting records of past warm intervals (i.e., high-$CO_2$ and elevated temperature climates) in order to better understand ice sheets and Southern Ocean

configurations under these conditions. For example, the early Pliocene (5-3 Ma) has been targeted because atmospheric $CO_2$ concentrations were similar to today's 400 ppmv concentrations (Foster and Rohling, 2013; Zhang et al., 2013). These studies have shown that early Pliocene Southern Ocean surface waters were warmer (i.e., between 2.5- > 4 ºC) than present and that the summer sea ice cover was greatly reduced, or even absent (Bohaty and Hardwood, 1998; Whitehead and Bohaty, 2003;

Escutia et al., 2009; Cook et al., 2013). They also record the periodic collapse of both the WAIS and EAIS marine-based margins (Naish et al., 2009; Pollard and DeConto, 2009; Cook et al., 2013; Reinardy et al., 2015; DeConto and Pollard, 2016). Foster and Rohling (2013) provide a sigmoidal relationship between eustatic sea-level and atmospheric $CO_2$ levels whereby sea levels stabilise at ~22 +/-12 m above present-day level, between about 400 ppm and 650 ppm, suggesting loss of the

Greenland Ice Sheet (6-7 m s.l.e.) and the marine-based West Antarctic Ice Sheet (+7 m s.l.e.). This implies that continental EAIS volumes remained relatively stable during these times, but experienced mass loss of some (or all) its marine–based margins, relative to the present day. With $CO_2$ concentrations at > 650 ppm they infer further increases in sea level, suggesting this as a threshold for initiating retreat of the terrestrial margins of EAIS. With sustained warming, $CO_2$ concentrations of

more than 650 ppmv are within the projections for this century (Solomon, 2007; Field et al., 2014). The last time the atmosphere is thought to have experienced $CO_2$ concentrations above 650 ppmv was during the Oligocene (23.03-33.9 Ma), when $CO_2$ values remained between 400 to ~750-800 ppm (Pagani et al., 2005; Beerling and Royer, 2011; Zhang et al., 2013).

Geological records of heavy isotope values ~2.5 ‰ and far field sea level records from passive margins during the Oligocene suggest that, following the continental-wide expansion of ice during the Eocene-Oligocene transition that culminated at the Oi-1 event (33.6 Ma), the Antarctic ice cover was at least ~50 % of the current volume (e.g., Kominz and Pekar, 2001; Zachos et al., 2001; Coxall et al., 2005; Pekar et al., 2006; Liebrand et al., 2011, 2017; Mudelsee et al., 2014). The early part of the Oligocene

records a significant $\delta^{18}O$ decreasing slope with high-latitude sites exhibiting a strong deglaciation/warming that persisted until ~32 Ma (Mudelsee et al., 2014). This was followed by seemingly stable conditions on Antarctica as evidenced by minimal $\delta^{18}O$ and Mg/Ca changes (Billups and Schrag, 2003; Lear et al., 2004; Mudelsee et al., 2014). A slight glaciation/cooling is recorded before 28 to ~27 Ma, which was followed by an up to 1 ‰ long-term decrease in the $\delta^{18}O$ isotope

records that was interpreted to result from the deglaciation of large parts of the Antarctic ice sheets during a significant warming trend in the late Oligocene (27-26 Ma) (Zachos et al., 2001a). Nevertheless, there are marked differences between the late Oligocene low $\delta^{18}O$ values recorded in Pacific, Indian and Atlantic Ocean sites (e.g., Pälike et al., 2006; Cramer et al., 2009; Liebrand et al., 2011; Mudelsee et al., 2014; Hauptvogel et al., 2017), and the sustained high $\delta^{18}O$ values recorded in

Southern Ocean sites (Pekar et al., 2006; Mudelsee et al., 2014). High $\delta^{18}O$ values in the Southern

Ocean sediments are in agreement with the ice proximal record recovered by the Cape Roberts Project (CRP) in the Ross Sea, which show the existence of glaciers/ice sheets at sea level (Barrett et al., 2007; Hauptvogel et al., 2017). Based on the study of the isotopic record in sediments from the Atlantic, the Indian and the equatorial Pacific, Pekar et al. (2006) explained this conundrum of a glaciated Antarctica, and varying intrabasinal $\delta^{18}O$ values with the coeval existence of two deep-water masses, one sourced from Antarctica and another, warmer bottom-water, sourced from lower latitudes. Superimposed on the above long-term swings in the $\delta^{18}O$ Oligocene record, fluctuations on timescales shorter than several Myr were identified in the high-resolution benthic $\delta^{13}C$ record from ODP 1218 (Pälike et al., 2006). These fluctuations in periods of 405 kyr and 1.2 Myr are related to Earth's orbital variations in eccentricity and obliquity, respectively and have been referred as the short-term "heartbeat" of the Oligocene climate (Pälike et al., 2006). Oligocene records close to Antarctica are needed to better resolve Antarctic ice sheet and paleoceanographic configurations at different time scales and under scenarios of increasing atmospheric $CO_2$ concentrations.

Integrated Ocean Drilling Program (IODP) Expedition 318 drilled a transect of sites across the eastern Wilkes Land margin at the seaward termination of the Wilkes Subglacial Basin (WSB) (Escutia et al., 2011; Escutia et al., 2014) (Fig. 1). Relatively good recovery (78.2 %) of late Oligocene (26-25 Ma) sediments from Site U1356 between 689.4 and 641.4 meters below sea floor (mbsf) provides an opportunity to study ice-sheet and ocean configurations during the late Oligocene and to relate them with other Antarctic and global records. In this paper, we present a new glacial-interglacial sedimentation and paleoceanographic model for the distal glaciomarine record of the Wilkes Land margin constructed on the basis of sedimentological data (visual core description, facies analysis, computed tomography images, and high-resolution scanning electron microscopy images), physical properties (i.e., magnetic susceptibility of the bulk sediment and grain density), and X-ray fluorescence data (XRF). We also provide insights into the configuration of the ice sheet in this sector of the east Antarctic margin and evidence for orbital forcing of the glaciomarine glacial-interglacial sedimentation at Site U1356.

## 2. Materials and Methods

### 2.1 Site U1356 description

Site U1356 (63º 18.6138'S, 135º 59.9376'E) is located at 3992 m water depth in front of the glaciated margin of the eastern Wilkes Land Coast of East Antarctica, and penetrated 1006 meters into the flank of a levee deposit in the transition between the lower continental rise and the abyssal plain (Escutia et al., 2011; Fig. 1). Overall recovery was 35% with sediments dated between the early Eocene and

Pliocene, but several intervals provide good stratigraphic control (Escutia et al., 2011; Tauxe et al., 2012). The Oligocene section was recovered between 895 and 430.8 mbsf, Cores U1356-95R-3 83 cm to U1356-46R. Our study focuses on the relatively high-recovery (78.2 %) interval within the late Oligocene, which spans from 689.4 to 641.4 mbsf (Cores U1356-72R to -68R). The sediments from this interval are part of shipboard lithostratigraphic Unit V, which is characterized by light greenish-grey,

strongly bioturbated claystones and micritic limestones interbedded with dark brown, sparsely bioturbated, parallel- and ripple-laminated claystones with minor cross-laminated interbeds (Escutia et al., 2011). The bioturbated and calcareous claystones and limestones were broadly interpreted to represent pelagic sedimentation superimposed on the background hemipelagic sedimentary input (Escutia et al., 2011). The laminated claystones and ripple cross-laminated sandstones were interpreted

to likely result from variations in bottom current strength and fine-grained terrigenous supply (Escutia et al., 2011). In addition, a notable absence of Ice Rafted Debris (IRD) (>250μm) in this interval relative to underlying and overlying strata was also recorded.

The late Oligocene depositional setting of Site U1356 was however different to that of today. The

stratigraphic evolution of the region testifies the progradation of the continental shelf taking place after continental ice sheet build-up during the Eocene-Oligocene Transition (EOT, 33.6 Ma; Eittreim et al., 1995; Escutia et al., 2005; 2014), which resulted in: 1) seismic and sedimentary facies on the continental rise becoming more proximal up-section (Hayes and Frakes, 1975; Escutia et el., 2000; 2005; 2014), and 2) high sedimentation rates during the Oligocene (Escutia et al., 2011; Tauxe et al.,

2012). In this context, the studied late Oligocene sediments from Site U1356 record distal continental rise deposition in an incipient/low-relief levee of a submarine channel. As progradation continued, a complex network of well-developed channels and high-relief levee systems developed on the continental rise (Escutia et al., 2000) from the latest Oligocene onwards.

Today, Site U1356 lies close to the Southern Boundary of the Antarctic Circumpolar Current, near the Antarctic Divergence at ~63ºS (Orsi, 1995; Bindoff, 2000) (Fig. 1). However, the paleolatitude of Site U1356 was around 58.5±2.5ºS (van Hinsbergen et al., 2015) during the late Oligocene, more northerly than today. Scher et al. (2008, 2015) reconstructed the position of the early Oligocene Antarctic Divergence to be located around 60ºS (Fig. 1), based on the distribution of terrigenous and biogenic

(calcareous and siliceous microfossils) sedimentation, Nd isotopes, and Al/Ti ratios through a core transect across Australian-Antarctic basin in the Southern Ocean. According to these interpretations Site U1356 lay far to the north of the Antarctic Divergence zone, and was closer to the Polar Front, during the Oligocene.

## 2.2 Age Model

The age model for Site U1356 was established on the basis of the magnetostratigraphic datums constrained by marine diatom, radiolaria, calcareous nannoplankton and dinocyst biostratigraphic control (Escutia et al., 2011; Tauxe et al., 2012; Bijl et al., 2018). The late Oligocene interval contains three magnetostratigraphic datums (Table 1): 1) Chron C8n.1n (o) between 643.70 and 643.65 mbsf (U1356-68R-2); 2) C8n.2n (y) between 652.60 and 652.55 mbsf (U1356-69R-2), and 3) C8n.2n (o) between 679.90 and 678.06 mbsf (U1356-71R). For this study, the age model by Tauxe et al. (2012), which was calibrated to the Gradstein 2004 Geological Time Scale, has been updated using the GPTS 2012 Astronomic Age Model (Vanderberghe et al. 2012). Based on this calibration, the age of sediments between 678.98 and 643.37 mbsf is 25.99 and 25.26 Ma, respectively (Fig. 2; Table 1).

## 2.3 Facies Analyses

Detailed facies analyses provide a stratigraphic framework on which we base our sedimentary processes and paleoenvironmental interpretations. Lithofacies are determined on the basis of detailed visual logging of the core during a visit to the IODP-Gulf Coast Repository (GCR), expanding on the lower resolution preliminary descriptions in Escutia et al. (2011). We logged the lithology, sedimentary texture (i.e., shape, size and distribution of particles) and structures with a focus on the contacts between the beds and on bioturbation at a mm to cm-scale resolution in cores expanding from 896 to 95.4 mbsf (Cores U1356-95R to -11R) (see Supplementary material S1 Fig. S1, S2). Physical properties data were measured during IODP Exp. 318 using the Whole-Round Multisensor Logger. Magnetic susceptibility measurements were taken at 2.5 cm intervals, and Natural gamma radiation (NGR) was measured every 10 cm (Escutia et al., 2011). In this paper, we focus on the interval between 689.4 and 641.4 mbsf that comprise cores 72R to 68R (Fig. 2).

X-ray Computed Tomography scans (CT-scans) measure changes in density and allow for analysis of fine-scale stratigraphic changes and internal structures of sedimentary deposits in a non-destructive manner (e.g., Duliu, 1999; St-Onge and Long, 2009; Van Daele et al., 2014; Fouinat et al., 2017). To further characterize the different facies in our cores, selected intervals of Core U1356-71R-6 (678.11 to 676.91 mbsf) and Core U1356-71R-2 (672.8 to 671.35 mbsf) were CT-scanned at the Kochi Core Center (KCC) (Japan), with the GE Medical systems LightSpeed Ultra 16. 2D scout (shooting conditions at 120Kv with 100mA, and 3D Helical image with 120Kv and 100mA and FOV=22.0). Image spatial resolution consists of 0.42 mm/pixel with 0.625 mm of slice thickness (voxel spatial resolution of 0.42 x 0.42 x 0.625 mm).

The type and composition of biogenic and terrigenous particles, particle size, and morphology of each lithofacies was characterized with a high-resolution scanning electron microscope (HR-SEM) at the Centro de Instrumentación Científica (University of Granada, Spain).

## 2.4 X-Ray Fluorescence (XRF) analyses

Detailed bulk-chemical composition records acquired by XRF core scanning allow accurate determination of sedimentological changes and the assessment of the contribution of the various components in the biogenic and lithogenic fraction in marine sediments (Croudace et al., 2006). This non-destructive method yields element intensities on the surface of split sediment cores and provides statistically significant data for major and minor elements (Richter et al., 2006; O'Regan et al. 2010, Wilhelms-Dick et al., 2012). The data are given as element intensities in total counts.

XRF core scanning measurements were collected every 2 cm down-core over a 1 cm$^2$ area with split size of 10 mm, a current of 0.2 mA (Al - Fe) and 1.5 mA (all other elements) respectively, and a sampling time of 20 seconds, directly at the split core surface of the archive half with XRF Core Scanner III at the MARUM − Center for Marine Environmental Sciences, University of Bremen, Germany. Prior to the scanning, cores were thermally equilibrated to room temperature, the surface was cleaned, flattened, and covered with 4 μm thin SPEXCerti Prep Ultralene1 foil to protect the sensor and prevent contamination during the scanning procedure. Scans were collected during three separate runs using generator settings of 10 kV for the elements Al, Si, S, K, Ca, Ti, Mn, Fe; 30 kV for elements such as Br, Rb, Zr, Mo, Pb; and 50 kV for Ba. The here reported data have been acquired by a Canberra X-PIPS Silicon Drift Detector (SDD; Model SXD 15C-150-500) with 150eV X-ray resolution, the Canberra Digital Spectrum Analyzer DAS 1000 and an Oxford Instruments 100W Neptune X-ray tube with rhodium (Rh) target material. Raw data spectra were processed by the Analysis of X-ray spectra by Iterative Least square software (WIN AXIL) package from Canberra Eurisys. Data points from disturbed intervals in the core face (i.e., slight fractures and cracks) were removed.

The light elements Al, Si, and K show large element variations (intra-element variations of 1 order of magnitude or more, Fig. 2). Similar variations have been previously described in sediment cores to indicate substantial analytical deviations due to physical sedimentary properties (i.e. Tjallingii and Röhl et al., 2007; Hennekam and de Lange 2012). Accordingly, for this study we have discarded the continuous records of Al, Si, and K and concentrated our interpretations on Al, Si and K values from the XRF analyses in discrete samples (see below). As Titanium (Ti) is restricted to the terrigenous phase in sediments and is inert to diagenetic processes (Calvert and Pedersen, 2007), we utilized Ti to

normalize other chemical elements for the terrigenous fraction. Linear correlation (r Pearson) above standardised values has been done in order to find statistical relationships among the variables.

In addition, we conducted measurements of a total of 50 major and minor trace elements in 25 discrete sediment samples collected at 0.4 and 1 m spacing to determine their chemical composition. For this, we used a Pioneer-Bruker X-Ray Fluorescence (XRF) spectrometer S4 at the Instituto Andaluz de Ciencias de la Tierra (CSIC-UGR) in Spain, equipped with a Rh tube (60 kV, 150 mA) using internal standards. The samples were prepared in a Vulcan 4Mfusion machine and the analyses performed using a standard-less spectrum sweep with the Spectraplus software.

## 2.5 Spectral Analyses

We selected key environmental indicators from XRF core scanner data and elemental ratios (i.e., Zr/Ba, Ba, Zr/Ti, Ca/Ti, MS) to conduct spectral analyses on the data from the interval between 689.4 to 641.4 mbsf (Cores U1356-72R to 68R). We performed evolutionary spectral and harmonic analysis on each dataset using Astrochron toolkit on the R software (Meyers, 2014). Detailed methodology is provided as supplementary information following the Astrochron code of Wanlu Fu et al. (2016). This method allows the detection of non-stationary spectra variability within the time series. The time series were analysed on the depth scale and then, applying the Frequency domain minimal tuning (Meyers et al., 2001), we converted spatial frequencies to sedimentation rates using an average period of 41 Kyr, to transform them to an age scale, with the basis of the already resolved age model. The Evolutionary Average Spectral Misfit method was then used to resolve unevenly sampled series and changing sedimentation rates (Meyers et al., 2012). This method is used to test a range of plausible timescales and simultaneously evaluate the reliability of the presence of astronomical cycles (Supplemental material S2).

## 3. Results

### 3.1 Sedimentary facies

The revised Oligocene facies log (Fig S1, S2), includes the high-recovery interval between 689.4 and 641.4 mbsf (Fig. 2). The integration of our lithofacies analyses, with physical properties (MS), CT-sans and HR-SEM analyses characterize an alternation between two main facies (Facies 1 and 2) (Figs. 2, 3, 4). Although these two facies were already visually identified shipboard, our analyses allow us a more detailed characterization and interpretation of the depositional environments and the processes involved in their development.

Facies 1 (F1) consists of slightly bioturbated greenish claystones with sparse (Fig. 3a) to common laminations (Figs. 2, 3a-f; Table 2). Laminae, as described on shipboard, vary from 0.1 to 1 cm thick and, based on non-quantitative smear slide observations, are composed of well-sorted silt to fine sand size quartz grains (Escutia et al., 2011). Laminations can be planar, wavy, with ripple-cross lamination structures (Escutia et al., 2011), and show faint internal truncation surfaces, mud offshoots, and internal erosional surfaces (Fig. 3a-f). HR-SEM analyses of the claystones show that the matrix is composed of clay-size particles and clay minerals (Fig. 3g, i). In addition, they show rare calcareous nannofossils that are partially dissolved (Fig. 3g, i). Authigenic carbonate crystals are also identified (Fig. 3i). Bioturbation in F1 is scarce, ichnofossils in the sediments are dominated mainly by *Chondrites* Fig. 3d). CT-scans also show the presence of *Skolithos*, with their vertical thin tubes filled with high-density material suggesting they are pyritized (Fig. 3b). Pyrite was also observed in shipboard smear slides in small abundances from the laminated facies in the studied interval (Escutia et al., 2011). Magnetic susceptibility values within the laminated facies are low, between 40-70 MS instrumental units (iu), with higher values when silt laminations are more abundant (Figs. 2, 4). Natural Gamma Ray (NGR) is anti-correlated with MS, with high values in F1 varying between 50 - 65 counts per second (cps) (Fig. 2).

Facies 2 (F2) is composed by light greenish grey strongly bioturbated claystones and silty claystones (Figs, 2, 3; Table 2) with variable carbonate content varying between 5-16% based on our XRF analyses. No primary structures are preserved due to the pervasive bioturbation (Fig. 3a-c). Burrows are backfilled with homogeneous coarse material (silt/fine sand). Different types of ichnofossils are present with *Planolites* and *Zoophycos* being the most abundant (Fig. 3a, b). HR-SEM images show: 1) silt-size grains containing quartz grains with conchoidal fractures in the corners and impact marks on the crystal faces, indicative of high-energy environments; and 2) biogenic carbonate consisting of moderately to poorly preserved coccoliths, which exhibit dissolution of their borders, and to a minor degree detrital carbonate grains (Fig. 3h - j). A total of 13 carbonate-rich layers have been observed within the studied interval F2, and they range in thickness from 10 to 110 cm. Facies 2 CT-scans images show an increase in density (i.e., gradation towards lighter colours in the scan) towards the top of each bioturbated interval (Fig. 3b). MS values are higher in F2 compared to F1. Values vary from 50-150 instrumental units (iu) and exhibit an inverse grading or a bigradational-like morphology (Fig. 2, 4), while NGR is inversely correlated with minimum values occurring in F2 (between 35-55 cps) (Fig. 2).

Contacts between the two facies are sharp and apparently non-erosive, with minimal omission surfaces or lags (Figs. 3,4). However, when bioturbation is present, gradual contacts in the transition from F1 to F2 also occur (Fig. 3b). Both sharp and transitional contacts are well imaged on the MS plots (Fig. 2).

In addition, where available, the CT-scan images confirm the shipboard and our own visual observations regarding the absence of outsized clasts and coarse sand grains in F1 and F2. Hauptvogel (2015) however, reports grains that are >150μm in size (fine sand fraction) as IRD. He argues that grains of that size could only reach Site U1356 through ice rafting given the distance of the site to shore, unless they were delivered by gravity flows. Thick and coarse-grained Mass Transport Deposits (MTDs) during the latest Oligocene at site U1356 (Escutia et al., 2011), argue for coarse material being delivered to the site by gravity flows. In addition, fine sand grains to gravel size clasts have been reported from channels on the lower continental rise off the Wilkes Land margin transported by gravity flows, including turbidity flows (Payne and Conolly, 1972; Escutia et al., 2000; Busetti et al., 2003). Given that during the late Oligocene, Site U1356 is located on a low-relief levee of a submarine channel, one can expect delivery of fine-grained sand and even coarser sediment to the site. In any case, even if some background IRD is present in our record, we argue it is minimal compared to elsewhere in the core.

## 3.2 Geochemistry

Down-core changes in the log ratios of various elements have been plotted against the facies log (Figs. 2, 4). In addition, in order to determine geochemical element associations we performed a Pearson correlation coefficient analysis of major elements on the whole XRF-scanner dataset (Table 3). This analysis highlights two main groups that are used as proxies for terrigenous (i.e., Zr, Ti, Rb, Ba) vs biogenic (i.e., Ca = carbonate) sedimentation.

Titanium (Ti), Zirconium (Zr), and Rubidium (Rb) are primarily derived from terrigenous sources, where Ti represents the background terrigenous input. During sediment transport Zr, Rb and Ti tend to become concentrated in particular grain-size fractions due to the varying resistance of the minerals in which these elements principally occur. Zr tends to become more concentrated in fine sand and coarse silt fractions, Ti in somewhat finer fractions and Rb principally in the clay-sized fraction (Veldkamp and Kroonenberg 1993; Dypvik and Harris 2001). The lack of correlation between Zr and Ti (Fig. 2; Table 3) implies that they are settled in different minerals and processes. The Zr/Rb ratio has been applied as a sediment grain-size proxy in marine records (Schneider, et al., 1997; Dypvik and Harris 2001; Croudace et al., 2006; Campagne et al., 2015). Zr/Al has been interpreted as an indicator for the accumulation of heavy minerals due to bottom currents (Bahr et al., 2014). In our cores, Zr/Rb and Zr/Ti ratios have a near identical variability downcore (Fig. 2). We utilize the high-amplitude Zr/Ti signal in our records as indicator for larger grain-size and current velocity (Fig. 2). The Zr/Ti ratio varies between 0.1 and 1 and exhibits maximum values within F2 showing an increasing upwards or

bigradational patterns (Fig. 2). Although minimum Zr values (cps) are found in F1, laminations with coarser-grained sediment within this claystone facies are also characterized by elevated Zr values similar to those in F2 (Figs. 3, 4; Table 3). The Zr/Ti pattern is positively correlated with magnetic susceptibility throughout the studied interval (Fig. 2).

The Zr/Ti, Zr/Rb and Zr/Ba ratios co-vary characterizing the laminations within F1 and the alternation between F1 and F2 by defining the contacts between them (Figs. 2, 4). They also mark the coarsening upwards or bigradational tendency in F2 (Fig. 4). Of the three ratios, the Zr/Ba ratio is the one that highlights these patterns best (Figs. 2, 4).

Barium (Ba) is present in marine sediments mainly in detrital plagioclase crystals and in the form of barite ($BaSO_4$; Tribovillard et al., 2006). In the studied sediments, Ba and Ti have a correlation factor of $r^2$=0.66 (Table 3), which is taken to indicate that Barium is predominantly present as a constituent of the continental terrigenous fraction and/or that biogenic barite was sorted by bottom currents. Ba has maximum values (10,000 total counts) at the base of F1 and decreases upwards in a saw-tooth pattern, reaching minimum concentrations within F2 (5,000 total counts) (Fig. 2; Table 3). The detrital fraction of Ba in the open ocean has been used in other studies as a tracer of shelf waters (Moore and Dymond, 1991; Abrahamsen et al., 2009; Roeske, 2011) and Ba record also is affected by current intensity in other depositional contourite systems (Bahr et al., 2014) preventing his use as paleoproductivity proxy in environments dominated by contour currents.

Variations in Ca, Mn, and Sr are strongly intercorrelated (Fig. 2) with $r^2$>0.87 (Table 3). Biogenic calcite precipitated by coccoliths and foraminifera have greater Sr concentration than inorganically precipitated calcite or dolomite (Hodell et al., 2008). The positive Ca and Sr correlation could therefore potentially be used to differentiate between terrigenous Ca sources (e.g. feldspars and clays) and biogenic carbonates (e.g. Richter et al., 2006, Foubert and Henriet, 2009, Rothwell and Croudace, 2015). Based on these observations, we interpret that Ca in our sediments is mainly of biogenic origin ($CaCO_3$). This interpretation is supported by HR-SEM images taken from carbonate-rich intervals of F2, which show abundant coccoliths (Fig. 3d). Peaks in Ca in our record (Fig. 2) coincide with the carbonate-rich layers listed in the previous section. Additional peaks in the record may indicate carbonate-rich layers that we have been unable to identify visually.

In order to estimate the $CaCO_3$ content continuously throughout the studied interval we use a calibration ($r^2_{U1356}$=0.81) between natural logarithm (ln) of Ca/Ti ratio (ln(Ca/Ti)) from the XRF core scanner data and the XRF discrete $CaCO_3$ measurements (weight %) from Site U1356 as applied in other studies (Zachos et al., 2004; Liebrand et al., 2016) (Fig. 5). "$CaCO_3$ est." is used throughout the text to refer to

carbonate content estimated by ln(Ca/Ti) ratio. CaCO$_3$ est. concentrations are generally low (between 0-16%). Carbonates are mostly present in F2, varying between 5-16 %, although small contents (from 0 to 5 %) can be seen in the intervals of F1 with scarce laminations (Fig. 4). CaCO$_3$ est. peaks in some intervals have a particular morphology producing a double peak in the beginning and/or the end of bioturbated F2 (Figs. 2, 4).

Mn(II) is soluble under anoxic conditions and precipitates as Mn(IV) oxyhydroxides under oxidising conditions (Tribovillard et al., 2006). Manganese is frequently remobilized to the sedimentary pore fluids under reducing conditions. Dissolved Mn can thus migrate in the sedimentary column and (re)precipitate when oxic conditions are encountered (Calvert and Pedersen, 1996). As such, large Mn enrichments primarily reflect changing oxygen levels at the sediment–water interface (Jaccard et al., 2016). The strong-correlated peaks of Mn and Ca (Fig. 2; Table 3) suggest that at least some of the Mn is present in the studied interval as Mn carbonates and/or Mn oxyhydroxides under oxic sediment-water interphase (Calvert and Pedersen, 1996; Calvert and Pedersen 2007; Tribovillard et al., 2006).

Br/Ti has been previously used as an indicator of organic matter in sediments (e.g., Agnihotri et al., 2008; Ziegler et al., 2008; Bahr et al., 2014). Br/Ti in our record shows generally low values (Fig. 2) most likely as the organic matter content in both facies types is relatively low (<0.5 %, Escutia et al., 2011). However, it exhibits some variability (0.01 to 0.05 Br/Ti ratio) within the two facies with higher ratio values in F1. Darker coloured sediments in F1 are in agreement with these higher Br/Ti values inside F1.

In addition to the elemental analyses of the XRF-scanned data, we use the detrital Al/Ti ratio in discrete XRF bulk sediment samples to reflect changes in terrigenous provenance (Kuhn and Diekmann, 2002; Scher et al., 2015). Al/Ti ratio varies between 17-21, with the highest values found within F1 and the lowest in F2 (Fig. 2).

**3.4 Spectral analysis**

To detect periodical signals, spectral analysis of time series was performed on the Zr/Ba and other elemental proxies (i.e., Ba, Zr/Ti, CaCO$_3$, Magnetic Susceptibility) using Astrochron R software (Meyers, 2014; Figs. 6; S3-10).

Multiple-taper spectral analysis (MTM) in Zr/Ba show a clear and statistically significant (>90%) cyclicity every 2m (0.5 cycles/m), and at 4.67m (0.21 cycles/m), and less significant one (>80%) at 1m (0.94 cycles/m) (Fig. S3). On the basis of a linearly calculated sedimentation rate between the two

extreme tie-points (Table 1), we obtained a sedimentation rate of approximately 5 cm/kyr. Within this sedimentation rate, the 0.5 cycles/m peak corresponds to the 41-kyr obliquity frequency; and the 0.21 and 0.94 cycles/m to the 95 and 21-kyr shorter eccentricity periods and precession frequencies, respectively.

After initial analysis, we ran an Evolutive Harmonic Analysis (EHA) (Astrochron (Meyers, 2014)) with 3 data tapers for the untuned Zr/Ba in depth domain with 2 cm resolution (Fig. S3). The statistical significance of spectral peaks was tested relative to the null hypothesis of a robust red noise background, AR(1) modelling of median smoothing, at a confidence level of 95% (Mann and Lees, 1996). Despite a short core gap in the middle of the time series, obliquity (41 kyr) dominates throughout the time series (Fig. 6). The sedimentation rates obtained by this method vary between 4.6 and 5.4 cm/kyr for the studied section, similar to those obtained with linearly calculated sedimentation rates. Additionally, the Nyquist frequency for Zr/Ba data is 1 m$^{-1}$ (0.5 kyr), which implies the site is sampled sufficiently to resolve precessional scale variations however, core gaps prevent identification of long eccentricity cycles (Fig. S6). Time series were anchored to the more robust paleomagnetic tie point in the U1356 age model, which is 25.99 Ma at 678.78 mbsf (Fig. S7).

Apart from obliquity, spectral analyses of the tuned age model reveal an alignment of the eccentricity and precession bands (Fig. 6, S8). For example, a marked cyclicity at the obliquity periods of 41 Kyr is seen at Ba and Zr/Ti (99% confidence) and also eccentricity at 100 kyr, and precession at 20kyr (95% confidence) (Fig. S9). We also observe coherent power above the 90% significance level at ~54 and ~29 ky periods, which are secondary components of obliquity. The anchored age model provides an unprecedented 500 yr resolution (2.5 cm sampling) of the data during the Late Oligocene. Orbital frequencies were tested in each core section individually in the Zr/Ba dataset in the depth scale in order to assure that cyclicity is not an artefact related to the gaps in the series (Fig. S10).

## 4. Discussion

Based on the integration of the facies characterized on the basis of sedimentological data (visual core description, facies analysis, CT-scans, HR-SEM), physical properties (magnetic susceptibility, NGR), and geochemical data (XRF), we provide for the late Oligocene interval (26 to 25 Ma): 1) a new glacial-interglacial sedimentation model for the distal glaciomarine record in the Wilkes Land margin dominated by bottom-current reworking of both, glacial and interglacial deposits; 2) insights into the configuration of the ice sheet in this sector of the east Antarctic margin; 3) changes in the

paleoceanographic glacial-interglacial configuration; and 4) evidence for orbital forcing of the glaciomarine glacial-interglacial sedimentation at Site U1356.

## 4.1 Glacial and interglacial contourite sedimentation off Wilkes Land

Laminated claystones (F1) from Site U1356 were originally interpreted by the shipboard science team to have formed during glacial times relating to variations in bottom current strength and fine-grained terrigenous supply. Conversely, the bioturbated claystones and micritic limestones (F2) were interpreted to result from mostly hemipelagic sedimentation during interglacial times (Escutia et al., 2011). Alternations between laminated deposits and bioturbated hemipelagic deposits, similar to those in F1

and F2, have been previously reported to characterize Pleistocene and Pliocene glacial-interglacial continental rise sedimentation, respectively, on this sector of the Wilkes Land margin (Escutia et al., 2003; Patterson et al., 2014). Gravity flows, mainly turbidity flows are the dominant process during glacial times resulting in laminated deposits. Interglacial sedimentation is dominated by hemipelagic deposition with higher opal and biogenic content (Escutia et al., 2003, Busetti et al., 2003). Erosion and

re-deposition of fine-grained sediment by bottom contour currents has also been reported as another important process during Pleistocene and Plio-Quaternary interglacials (Escutia et al., 2002; Escutia et al., 2003, Busetti et al., 2003).

       Despite being sparse, the occurrence of bioturbation in our laminated sediments in F1, which slightly

affects both claystones and silt laminations, indicates slow and continuous sedimentation. This is not consistent with instantaneous turbidite deposition, which would be expected at the Site U1356 located on the left low-relief levee of a contiguous channel during the late Oligocene. It is however consistent with fine-grained turbidite overbank deposits being consequently entrained by bottom currents. Silt layer sedimentary structures similar to those described by Rebesco et al. (2008, 2014) indicate that there

is current reworking of the sediments. For example, silt layers can be continuous or discontinuous with wavy and irregular morphologies, and within layers, sedimentary structures such as cross-laminations are common (Fig. 3c-f). Within the cross laminae, mud offshoots and internal erosional surfaces are distinctive features of fluctuating currents where successive traction and suspension events are super-imposed, indicating bottom-currents sedimentation as the principal process for the F1 laminated

claystones (Shanmugam et al., 1993; Stow et al., 2002). Based on these observations, we interpret F1 as glacial laminated muddy contourites following the classification of Stow and Faugères (2008). The F1 sedimentary structures suggest bottom-currents with fluctuating intensities, that result in laminations and internal structures forming during peak current velocities (Lucchi and Rebesco, 2007; Martín-Chivelet et al. 2008; Rebesco et al., 2014). Laminated, fossil-barren, glaciogenic deposits, consistent

with those of Facies F1, have been observed on younger sedimentary sections in glaciated margins and

interpreted as contour current modified turbidite deposits and as muddy contourites (Anderson et al., 1979; Mackensen et al., 1989; Grobe and Mackensen, 1992; Pudsey, 1992; Gilbert et al., 1998; Pudsey and Howe, 1998; Pudsey and Camerlenghi, 1998; Anderson, 1999; Williams and Handwerger, 2005; Lucchi and Rebesco, 2007, Escutia et al., 2009). This particular type of contourite facies is associated with glaciomarine deposition during times of glacial advance, and has been interpreted to result from unusual, climate-related, environmental conditions of suppressed primary productivity and oxygen-poor deep-waters (Lucchi and Rebesco, 2007).

Bioturbated sediments in F2 were previously interpreted as interglacial hemipelagic deposits (Escutia et al., 2011). In this study, we interpret F2 as hemipelagic and overbank deposits reworked by bottom-currents. The coarser grain-size in F2 compared to F1 (silty-clay matrix as seen in HR-SEM Fig. 3g-j), the distribution of heavy minerals as indicated by the Zr/Ba, and the elevated values of the magnetic susceptibility record with a bigradational pattern within the facies (Figs. 2,4), support the notion that interglacial sediments of F2 have been heavily modified by bottom currents. Hemipelagic sediments are expected to be homogeneous in terms of grain-size and grading is not expected. Current winnowing of hemipelagic deposits and removal of the fine-grained fraction can produce the higher accumulation of heavy (indicated by the Zr) and ferromagnetic (indicated by MS) minerals observed in F2 compared to F1 (Fig. 2; Table 2). High MS values result from stronger bottom currents deposition and/or increased terrigenous input (e.g., Pudsey et al., 2000; Hepp et al., 2007). Also, bigradational trends have been previously described in contourite sediments and interpreted to record an increase followed by a decrease in the current velocities (e.g., Martín-Chivelet et al., 2008). The bigradational patterns in the Zr/Ba and MS plots (Figs. 2,4) are therefore interpreted to depict a constant and smooth increase followed by a decrease in current velocity with little gradual changes in flow strength. In addition, the presence of grains of quartz with conchoidal fractures and reworked coccolitospheres with signs of dissolution (Fig., 3h,j) support the reworking of background hemipelagic and turbidite overbank sediments by bottom currents in a high-energy environment (Damiani et al., 2006). Following the classification by Stow and Faugères (2008), we interpret that F2 has more silty massive contourites resulting from higher and more constant bottom current velocity compared to F1.

Transitions between the F1 and F2 facies are characterized by glacial-to-interglacial contacts that may be sharp or diffuse due to bioturbation, and characterized by a gradual change in physical and geochemical sediment parameters (Figs., 3, 4; Table 3). Interglacial-to-glacial contacts (F2 to F1), on the other hand, are characterized by an apparently non-erosional sharp lithological boundary. The sharp lithological boundaries between interglacial to glacial transitions can be explained by maximum current intensities achieved at the end of the interglacials (Shanmugam, 2008; Rebesco et al., 2014).

## 4.2. Ice sheet configuration during the warm late Oligocene

Early Oligocene and post-Mid Miocene climate transition sediments from Site U1356 contain granule and larger clasts (>2mm) interpreted as ice rafted debris (IRD; Escutia et al., 2011; Sangiorgi et al., 2018; Fig. S1). In addition, dinocyst assemblages indicate the presence of sea ice (Houben et al., 2013). Based on this, one could expect the site to be within the reach of icebergs calving from an expanded ice sheet grounded at the coast or beyond in the late Oligocene. This is supported by Pliocene-Pleistocene sedimentary sections in adjacent continental rise sites containing IRD (Escutia et al., 2011; Patterson et al., 2014). In addition to the paucity of IRD in our studied interval, the absence of sea ice-loving species *Selenopemphix antarctica* and common to abundant gonyaulacoid phototrophic dinocysts, suggest warm-temperate surface waters (Bijl et al., submitted, this volume). A sea ice-free scenario during the late Oligocene is also supported by elevated sea surface temperatures (i.e., average summer temperatures are ~19°C) based on biomarker sea surface temperatures (TEX$_{86}$ data in Hartman et al., submitted, this volume). Furthermore, the presence of *in situ* terrestrial palynomorphs suggests that during the late Oligocene margins nearby were in part free of ice sheets and covered by a cool-temperate vegetation with trees and shrubs (Salzmann et al., 2016, Strother et al., 2017). All these observations suggest a reduced ice sheet and partly ice-free margins in the Wilkes margin during the late Oligocene.

These observations are consistent with the iceberg survivability modelling in the Southern Ocean for the warm Pliocene intervals, which shows the distance that icebergs could travel before melting was significantly reduced (Cook et al., 2014). Warm Pliocene summer sea surface temperatures up to 6°C warmer than today during interglacials and prolonged Pliocene warm intervals have been reported in the Ross Sea (e.g., Naish et al., 2009; McKay et al., 2012) and other locations around Antarctica (Whitehead and Bohaty, 2003; Whitehead et al., 2005; Escutia et al., 2009; Bart and Iwai, 2012). Contrary to what we observe in our late Oligocene record and in the Miocene Climatic Optimum (Sangiorgi et al., 2018), abundant IRD were delivered to continental rise sites adjacent to Site U1356 during the warm Pliocene (Escutia et al., 2011; Patterson et al., 2014). This was interpreted by Cook et al. (2017) to suggest that a considerable number of icebergs (iceberg armadas) had to be produced in order to reach the site under these warm Pliocene conditions. We argue that the lack of IRD delivery to site U1356 during the studied warm late Oligocene interval can result from the different Wilkes Subglacial Basin (WSB) late Oligocene paleotopographic setting. Paleotopographic reconstructions from 34 Ma ago (Wilson et al., 2012) and the early Miocene (Gasson et al., 2016), show the WSB to be an area of lowlands and shallow seas in contrast to the over-deepened marine basin that it is today (Fretwell et al., 2013). This paleotopographic configuration would have precluded widespread marine ice sheet instability during the Oligocene. This difference is important, as an ice sheet grounded on an

overdeepened continental shelf can experience marine ice sheet instability, a runaway process relating to ice sheet retreat across a reverse slope continental shelf (Weertman 1974), which is proposed to be a driver for retreat of the EAIS in the WSB during the warm Pliocene (Cook et al., 2013). Conversely, a shallower continental shelf allows for the potential expansion of grounded ice sheets into the marine margin during warmer-than-present climates (Wilson et al. 2012), and thus direct records are required to assess the climate threshold for such an advance.

In comparison to the distal U1356 Wilkes Land margin record, the Ross Sea Embayment ice proximal sediments obtained by the Cape Roberts Project (CRP) contain Oligocene to Early Miocene palynomorphs, foraminifera and clay assemblages that point to a progressive decrease in fresh meltwater, cooling and intensifying glacial conditions (Leckie and Webb, 1983; Hannah et al., 2000; 2001; Raine and Askin, 2001; Thorn, 2001; Ehrmann et al., 2005; Barrett, 2007). Therefore, the coastal CRP sediment record does not support a significant loss of ice or warming during the late Oligocene (Barrett, 2007). The high sedimentation rates during the late Oligocene-early Miocene recorded at Deep Sea Drilling Project (DSDP) Site 270 were interpreted to reflect turbid plumes of glaciomarine sediments derived from polythermal-style glaciers or ice sheets that were calving into an open Ross Sea, without an ice shelf (Kemp and Barrett, 1975). In addition, seismic data indicate that during the late-mid Oligocene widespread expansion of a marine-based ice sheet onto the outer Ross Sea shelf did not take place but instead glaciers and ice caps drained from local highs and advanced only into shallow marine areas, rather than whole-scale marine ice sheet advance (Brancolini et al., 1995; De Santis et al., 1995; Bart and De Santis, 2012).

Combined, these evidences suggest that during the late Oligocene marine-terminating glaciers, ice caps and glaciers persisted along the Transantarctic Mountain front reaching the Ross Sea coastal areas, but may have been more confined within a warmer WSB margin. This is also supported by vegetation reconstructions derived from fossil pollen from both margins, which indicate for the middle Miocene and Late Oligocene higher terrestrial temperatures and more tree taxa at Wilkes Land (Salzmann et al., 2016; Sangiorgi et al., 2018) than the Ross Sea (Askin and Raine, 2000; Prebble et al., 2006). This is consistent with the ice sheet modelled configuration for Miocene topographies with $CO_2$ scenarios of 500-840 ppm (Gasson et al., 2016; Levy et al 2016; Fig. 7).

## 4.3 Paleoceanographic implications

Sediment physical properties and geochemical signatures of F1 and F2 are here related to changes in bottom water-sediment interphase oxygenation/ventilation during successive glacial and interglacial periods (Table 2). We interpret that these changes are linked to shifts in water-masses driven by a north-

south displacement of the position of the westerlies, and associated changes in the intensity of frontal mixing or location of the Polar Front and Antarctic Divergence (Fig. 7). Based on our observations, we propose a model to explain the interpreted changes in bottom-water conditions at Site U1356 during successive glacial and interglacial times (Fig. 7).

### 4.3.1. Glacial paleoceanographic configuration

The *Chondrites*-like bioturbation with pyrite infilling the tubes of *Skolithos* within F1 (Fig. 3b, d) has previously been reported to characterize low-oxygen conditions at the water-sediment interphase (Bromley and Ekdale, 1984). In addition, pyritized diatoms are present throughout the Oligocene section of this site, but are found preferentially inside F1. The presence of pyritized diatoms was interpreted during Expedition 318 to indicate a prolific production and syn-sedimentary diagenesis in a restricted circulation (low oxygen) environment, mainly during glacial periods (Escutia et al., 2011). Reducing conditions in the sediment also help to preserve primary sedimentary structures of the silt layers in F1 because bioturbation is limited. Higher amounts of organic matter in F1 compared to F2 are suggested by increased values of the Br/Ti ratio (Fig. 2). The higher organic content most likely produces a poorly ventilated environment with near reducing conditions at the water-sediment interphase, where pyrite can precipitate (Tribovillard et al., 2006). In spite of this, total oxygen depletion did not occur as indicated by the palynomorphs good preservation within F1 (Bijl et al., submitted, this volume).

Low MS values such as those recorded within F1 (Fig.4; Table 2) have been reported around Antarctica and attributed to magnetic minerals dissolution caused by dilution and/or primary diagenesis effects on the sediments due to the higher concentration in organic matter or to changing redox conditions (Korff et al., 2016). Several authors have postulated that oxygen-depleted Antarctic Bottom Water (AABW) occupying the abyssal zones of the oceans can change the redox conditions in the sediment, trapping and preserving dissolved and particulate organic matter and, consequently reducing and dissolving both, biogenic and detrital magnetite (Florindo et al., 2003; Hepp et al., 2009; Korff et al., 2016). At present, Site U1356 is influenced by AABW forming in the adjacent Wilkes Land shelf (Orsi et al., 1999; Fukamachi et al., 2000) and in the Ross Sea spilling over to the Wilkes Land continental shelf (Fukamachi et al., 2010) (Fig. 1). Our records suggest a reduced continental ice-sheet in the eastern Wilkes Land margin and reduced sea ice presence compared to today (Bijl et al., submitted, this volume). Under these conditions, bottom water formation and downwelling can still occur (with or without presence of sea ice) as a result of density contrasts related to seasonal changes in surface water temperature and salinity (Huber and Sloan, 2001; Otto-Bliesner et al., 2002). Moreover, stable Nd

isotopic composition in Eocene-Oligocene sediments from Site U1356 is consistent with modern day formation of bottom water from Adélie Land, as reported by Huck et al (2017).

Our evidence above points to deposition of F1 during glacial cycles under poorly-ventilated, low-oxygenation conditions at the water-sediment interface (Fig. 7a). We postulate, that during glacial periods, westerly winds and surface oceanic fronts migrate towards the equator, generating a more stratified ocean and reduced upwelling closer to the margin, with sporadic and fluctuating currents (Fig. 7a). Records of the Last Glacial Maximum show that this northward migration results in a weakening of the upwelling of the Circumpolar Deep Water (CDW) (Govin et al., 2009), increasing stratification and reduced mixing of water masses also due to an enhanced sea ice formation, not seen during the late Oligocene.

### 4.3.2. Interglacial paleoceanographic configurations

We differentiate two interglacial paleoceanographic configurations based on the presence of some intervals of micritic limestone with calcareous nannofossils.

In general, the higher degree of bioturbation in F2 with no primary structures preserved and the ichnofacies association (i.e., *Planolites* and *Zoophycos*), suggest a more oxygenated environment in comparison with F1. This is supported by the covariance of Mn and $CaCO_3$ est. (Fig 4) where Mn enrichments can be interpreted as redox changes variations (Calvert and Pedersen, 2007; Jaccard et al., 2016). More oxygenated conditions during interglacial periods can be achieved under more ventilated and mixed water masses, with enhanced current velocities. Enhanced currents during deposition of F2 are interpreted based on coarser grain size, and the increased accumulation of heavy and ferromagnetic minerals as indicated by the high values of the Zr/Ti ratio and MS within F2 (Figs. 2,4). The bigradational pattern of the Zr/Ba and the MS (Fig. 4) is also interpreted to record an increase followed by a decrease in current velocities within F2.

The intervals of micritic limestone within F2 have calcareous nannofossils preserved (Fig 3d). The productivity of calcareous nannofossils and the later preservation of these coccoliths in the sediment indicate specific geochemical conditions enabling carbonate deposition and preservation. Although today nannoplankton is abundant in surface waters at the Antarctic Divergence (Eynaud et al., 1999), these rarely deposit on the deep ocean floor because of corrosive bottom waters, which dissolve calcareous rain. A number of studies in other areas of the Antarctic margin and the Southern Ocean have correlated the presence of calcareous nannofossils with the presence of temperate north component water masses (North Atlantic Deep Water-like, NADW) that intrude close to the Antarctic continent

and influence the Southern Ocean during the late Oligocene (e.g., Nelson and Cooke, 2001; Pekar et al., 2006; Villa and Persico, 2006; Scher and Martin, 2008), the Miocene (DeCesare et al., 2013; Sangiorgi et al., 2018), and during more recent times such as the Quaternary (Diekman, 2007; Kemp et al., 2010; Villa et al., 2012).

The more oxygenated and ventilated conditions in our records suggest enhanced mixing of the water masses (Fig. 7b-c). We postulate that during interglacials westerly winds and the Polar Front are shifted south and become more aligned. Under these conditions, upwelling of deep waters is likely promoted, facilitating the mixing and oxygenation of surface waters that form the precursor to bottom water.
Similar process has been reported for the Holocene by Peck et al. (2015). Such a process would also generate increased geostrophic current velocities of the bottom water mass, supported by the coarser grain size and heavy mineral concentrations in the bioturbated F2 facies.

Similar to what is occurring under the present warming, bottom water formation during interglacials is
likely fresher and less dense due to enhanced freshwater runoff from surface and subglacial melt of the continental ice sheet (Wijk and Rintoul, 2104). Today, a reduction in the volume of the AABW is compensated by the expansion of the Circumpolar Deep Water (CDW) (Wijk and Rintoul, 2014), which forms by mixing of abyssal, deep, and intermediate water masses, including the AABW and the NADW (Johnson, 2008). We hypothesize, that during warmer interglacials, the influence of more northern-
sourced water masses into the proto-CDW, relative to Antarctic-sourced (Fig. 7c), could enable carbonate productivity and preservation of coccolitosphere remains, seen at least 13 occasions in our record. These data are also in agreement with the $\delta^{13}C$ global isotopes oscillations between 26 and 25 Ma (Cramer et al., 2009; Liebrand et al., 2017), that suggest low values for an AABW and high $\delta^{13}C$ values for a NADW, that may represent the different oceanic primary production and ventilation rates,
as proposed in this work. In addition, $\delta^{13}C$ records in the Atlantic show systematic offsets to lower values toward a North Atlantic signal for most of the late Oligocene to early Miocene. These data suggest the influence of two distinct deep-water sources: cooler southern component water and warmer northern component water (Billups et al., 2002; Pekar et al., 2006; Liebrand et al., 2011). In addition, the increased presence of North Component Deep waters influencing this sector of the eastern Wilkes
Land margin could be related with a slowdown of the southern limb of the overturning circulation.

**4.4 Orbital forcing and Glacial and Interglacial cyclicity**

The first spectral analysis on late Oligocene sediments from the eastern Wilkes Land margin at Site U1356 shows that glacial-interglacial cycles, resulting in changes in the oceanic configuration off
Wilkes Land, are paced with variations in Earth's orbit and seasonal insolation. Although the data is

somewhat discontinuous due to gaps in our record, it clearly shows that the glacial-interglacial cyclicity (every 2 m or 41 kyr) discussed above has a persistent obliquity pacing throughout the studied late Oligocene interval (26-25 Ma) in the Wilkes Land. Consequently, this obliquity-paced cyclicity modulates the amount of deep-water production in the Southern Ocean, and exerts a major control on oceanic configuration and current strength. Bottom current velocity fluctuations and ventilation of bottom sediments respond to the forcings applied by the strength of the Southern Hemisphere westerlies, the position of the PF respect to the site, and consequently by the water mass occupying the bottom of the basin at each time. In addition to obliquity, precession is also present, which implies a dynamic response of the EAIS and offshore oceanic water masses to orbital forcing.

East Antarctic ice volume fluctuations at orbital periodicities in the obliquity band in the Wilkes Land margin have been previously reported from early warm Pliocene (3-5 Ma) sediments obtained from Site U1361 (Patterson et al., 2014). In the Ross Sea, cyclicity in sediments collected by the CRP from the late Oligocene, the late Miocene and the early warm Pliocene period was also paced by obliquity (Naish et al., 2001; McKay et al., 2009; Naish et al., 2009). Similar orbital variability in the deep-water circulation patterns have also been inferred to have occurred with the growth of the EAIS during the middle Miocene between 15.5 to 12.5 Ma (Hall et al., 2003). In addition, other studies have linked changes in Atlantic meridional overturning (Lisiecki et al., 2008; Scher et al., 2015) and Antarctic circumpolar ocean circulation (Toggweiler et al., 2008) to obliquity forcing. An interglacial mechanism has been proposed whereby the southward expansion of westerly winds and associated Ekman transport is compensated for by enhanced upwelling of warmer, $CO_2$-rich CDW (Toggweiler et al., 2008), which also promotes atmospheric warming. In the equatorial Pacific, Pälike et al. (2006) also report strong obliquity in the benthic $\delta^{13}C$ isotopic record between 26-25 Myr, implying that changes in the carbon cycle (pacing glacial /interglacial periods) are triggered in the high southern latitudes and transferred to the global deep-ocean through the bottom water masses.

## 5. Conclusions

Our study provides new insights regarding Antarctic ice sheet and paleoceanographic configurations that prevailed in the eastern Wilkes Land margin between 26 and 25 Ma. Sediments at IODP Site U1356 during this interval are characterized by the alternation between two main facies (F1 and F2), that are dominated by reworking by bottom-currents with varying intensities of glacial-interglacial gravity flows and hemipelagic deposits. Claystones with silty laminations (F1) are interpreted to represent fluctuating bottom current intensities during glacial periods. Massive bioturbated silty clays and micritic limestones with coccoliths (F2) are interpreted as interglacial deposits and record maximum velocities of bottom-currents at this site. The lack of iceberg rafted debris (IRD), the absence

of sea ice, elevated sea surface temperatures throughout the studied interval, and reconstructions of cool-temperate vegetation suggest that reduced glaciers or ice caps occupied the topographic highs and lowlands of the now overdeepened Wilkes Subglacial Basin between 26 and 25 Ma and that iceberg calving was only a background process during this time due to the lack of marine terminating ice sheets.

Glacial sediments record poorly ventilated, low-oxygenation conditions at the water-sediment interface that we postulate result when westerly winds and surface oceanic fronts migrate towards the equator and overturning is reduced near the Antarctic margin. During interglacial times, more oxygenated and better ventilated conditions are inferred to have prevailed which would act to enhance mixing of the water masses with increased current velocities. We postulate that during interglacials, westerly winds shifted south and became more aligned with the Antarctic Divergence and Polar Fronts, which promoted upwelling of deep waters and facilitated the mixing and oxygenation of bottom waters. Micritic limestone intervals within interglacial F2, record warmer paleoclimatic conditions when the influence of more northern-sourced water masses into the proto-CDW, relative to Antarctic-sourced (Fig. 7c), could enable carbonate productivity and preservation of coccolitosphere remains. Preservation of carbonate in some F2 intervals supports previous paleoceanographic studies that consider at least a two-layer ocean with an Antarctic Bottom Water (undersaturated with respect to calcium carbonate), and a proto-CDW with a greater influence of warmer Northern Component Water mass (NADW-like) to reconcile intra-basinal differences in $\delta^{18}O$ values (Pekar et al., 2006). Based on the number of carbonate-rich layers, warmer NADW-like waters reached the site at least 13 times during the studied interval.

Spectral analysis on late Oligocene sediments from the eastern Wilkes Land margin reveal that glacial-interglacial paleoceanographic changes during the late Oligocene are regulated primarily by obliquity, although frequencies in the eccentricity and precession band are also recorded. However, as we do not have a measure of ice dynamics during this time (e.g. ice rafted debris), the orbital response of terrestrial ice in the Wilkes Land Basin remains ambiguous, beyond what is inferred from the deep-sea isotope record.

**Acknowledgements**

This research used samples and data provided by the Integrated Ocean Drilling Program, now the International Ocean Discovery Program (IODP). We thank the staff onboard IODP Exp. 318 and at the Gulf Coast, the Bremen and the Kochi IODP Core Repositories for assistance in core handling and shipping. We thank Vera Lukies (MARUM) for technical

support with XRF core scanning and Shizu Yanagimoto (KOCHI) for technical support with CT-scans. We also thank the constructive comments of an anonymous reviewer and Dr. Steven Pekar that have helped to improve this manuscript. Funding for this research is provided by the Spanish Ministerio de Economía y Competitividad (Grants CTM 2011-24079 and CTM2014-60451-C2-1-P) co-funded by the European Union through FEDER funds. UR thanks the *Deutsche Forschungsgemeinschaft* (DFG) (RO 1113/6). PKB, FS and JDH acknowledges funding through NWO polar programme grant no 866.10.110; PKB acknowledges funding through NWO-VENI grant no 863.13.002. US acknowledges funding received from the Natural Environment Research Council (NERC grant NE/H000984/1).

**Author contributions**

CE and AS designed the research. PKB, JH, FS and HB, provided insights regarding biomarker-based sea surface temperatures and sea ice conditions based on dinocysts. UR provided XRF core-scanning data. FJJE and UR provided geochemical input. CHN and RM provided input with sedimentary and facies interpretations. MI provided the CT-Scans data. JAF provided input in the paleoceanographic interpretations. DE and ALQ provided Antarctic overview and petrographic input. SR and US provided palynology insights. AS and CE wrote the paper with input from all co-authors.

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

**Table 1: Age model by Tauxe et al., (2012) and transformed ages to GPTS 2012**

| Core Section Site U1356 Exp. 318 | Top depth (mbsf) | Bottom depth (mbsf) | Depth used (m) | GPTS 2004 (Myr) (Tauxe et al., 2012) | GPTS 2012 (Myr) | Chron |
|---|---|---|---|---|---|---|
| 68R-2 | 643.10 | 643.65 | 643.37 | 25.444 | 25.260 | C8n.1n (o) |
| 69R-2 | 652.55 | 652.60 | 652.57 | 25.492 | 25.300 | C8n.2n (y) |
| 71R-6 | 678.06 | 679.90 | 678.98 | 26.154 | 25.990 | C8n.2n (o) |

**Table 2: Types of facies differentiated by physical, geochemical, and biological character and their interpretation in terms of sedimentary processes and paleoclimate.**

| | | Facies 1 (F1) | Facies 2 (F2) |
|---|---|---|---|
| **Lithological description** | | Bioturbated green claystones with thin silt laminae with planar and cross-bedded laminations | Highly bioturbated, thicker pale-brown, silty-claystones |
| **Contacts** | **Top** | Gradual, bioturbated | Sharp |
| | **Bottom** | Sharp | Gradual, bioturbated |
| **Bioturbation** | | Sparse bioturbation. Primary structures preserved | Strong bioturbated. Massive. No primary structures preserved |
| **Nannos** | | Barren to rare | Barren to variable abundance and preservation |
| **IRD** | | No | No |
| **Magnetic susceptibility (MS)** | | Low in claystones and high in silty laminations | High |
| **XRF-Scanner elements concentration** | **Zr** | Low in claystones and high in silty laminations | High, (max. values on top) |
| | **Ba** | High, (max. values on bottom) | Low |
| | **Ca** | No | Variable, low to high |
| **Formation process** | | Bottom currents of fluctuating intensities | Bottom currents with higher velocity and constant flux |
| **Facies interpretation** | | Cold periods. Supply of terrigenous by density current flows, reworked by bottom currents. | Well-oxygenated deep-sea sedimentation. Warm periods with reworking of sediments by bottom currents |


**Table 3: R Pearson Linear correlation between XRF-scanner elements.**

|     | MS | S | Ca | Ti | Mn | Fe | Br | Rb | Zr | Sr |
|-----|------|------|------|------|------|------|------|------|------|------|
| S | -0.214 | | | | | | | | | |
| Ca | 0.226 | -0.122 | | | | | | | | |
| Ti | -0.212 | 0.620 | -0.290 | | | | | | | |
| Mn | 0.151 | -0.121 | 0.858 | -0.246 | | | | | | |
| Fe | 0.0419 | 0.0449 | -0.396 | 0.510 | -0.324 | | | | | |
| Br | -0.297 | 0.111 | -0.438 | 0.118 | -0.363 | 0.056 | | | | |
| Rb | -0.282 | 0.036 | -0.576 | 0.286 | -0.489 | 0.455 | 0.493 | | | |
| Zr | 0.480 | -0.164 | -0.036 | -0.099 | -0.058 | -0.055 | 0.102 | 0.067 | | |
| Sr | 0.186 | 0.006 | 0.871 | -0.074 | 0.677 | -0.345 | -0.303 | -0.515 | 0.040 | |
| Ba | -0.290 | 0.339 | -0.234 | 0.662 | -0.210 | 0.354 | 0.343 | 0.402 | 0.018 | 0.039 |








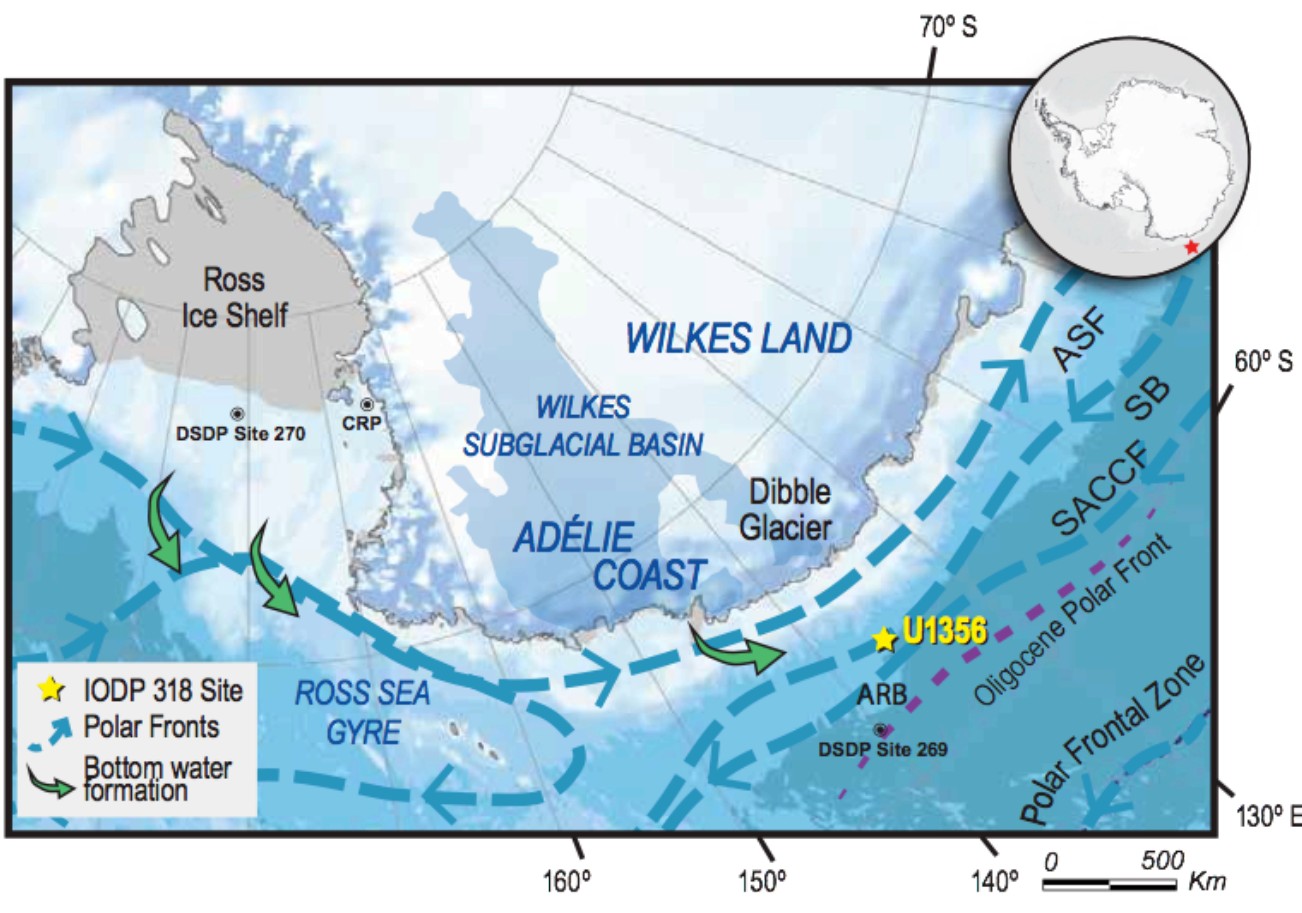



**Fig. 1:** Location of IODP 318 Site U1356 (Escutia et al., 2010) on the Adélie coast continental rise. Bed
topography from IBSCO2 (Arndt, JE et al., 2013). Schematic position of the different water masses at present
and locations of Antarctic Bottom Water formation (Orsi, 1995) are indicated. The position of the Oligocene
Polar Front (Scher et al., 2015) is also shown. ASF: Antarctic Slope Front; SB: Southern Boundary; SACCF:
Southern Antarctic Counter Current Front; ARB: Adélie Rift Block.


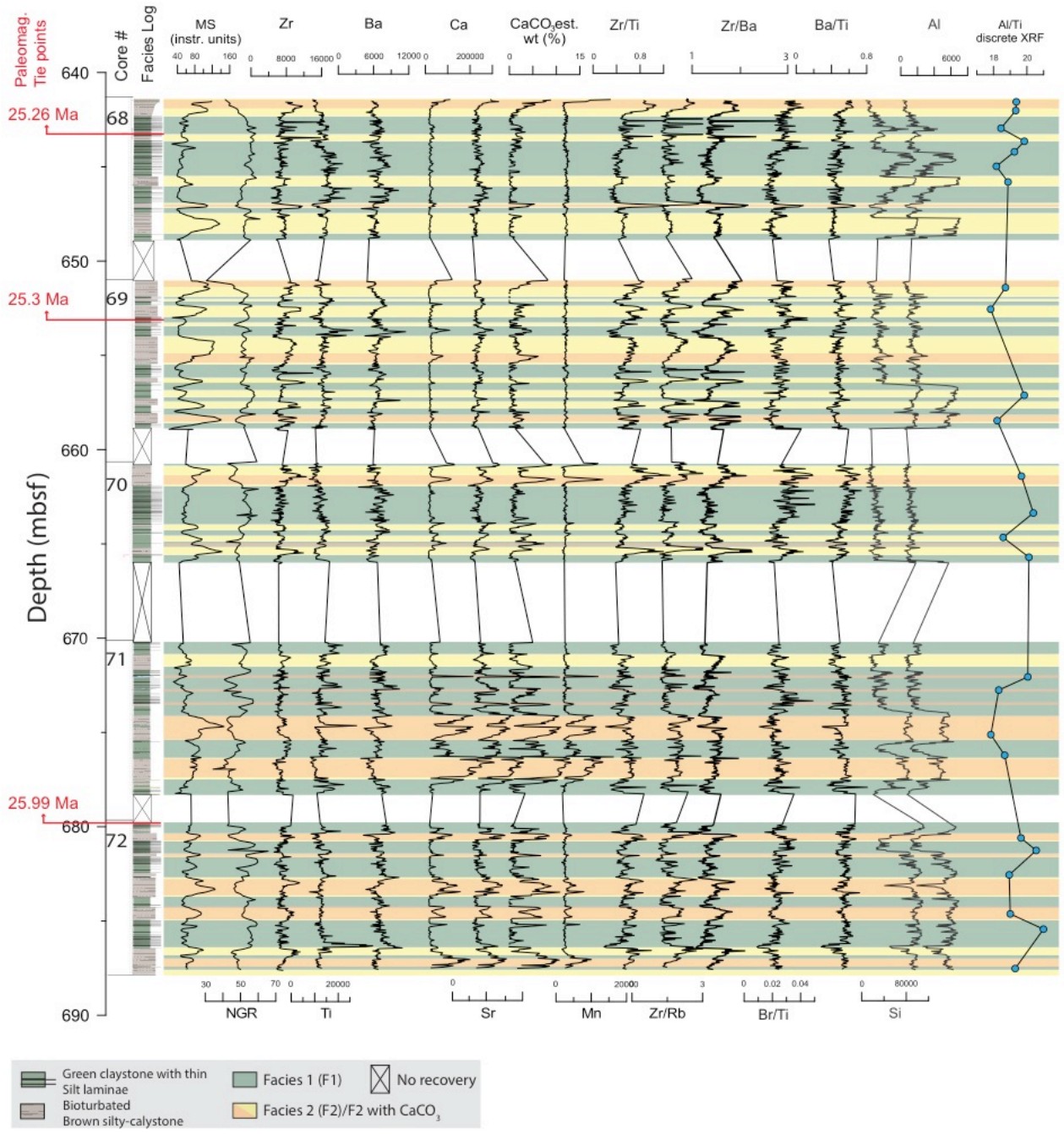

**Fig. 2:** Magnetic susceptibility (MS) and natural gamma radiation (NGR) physical properties, and selected X-Ray Fluorescence (XRF) data (in total counts) and elemental rations plotted against the new detailed U1356 facies log between 689.4 and 641.4 mbsf.

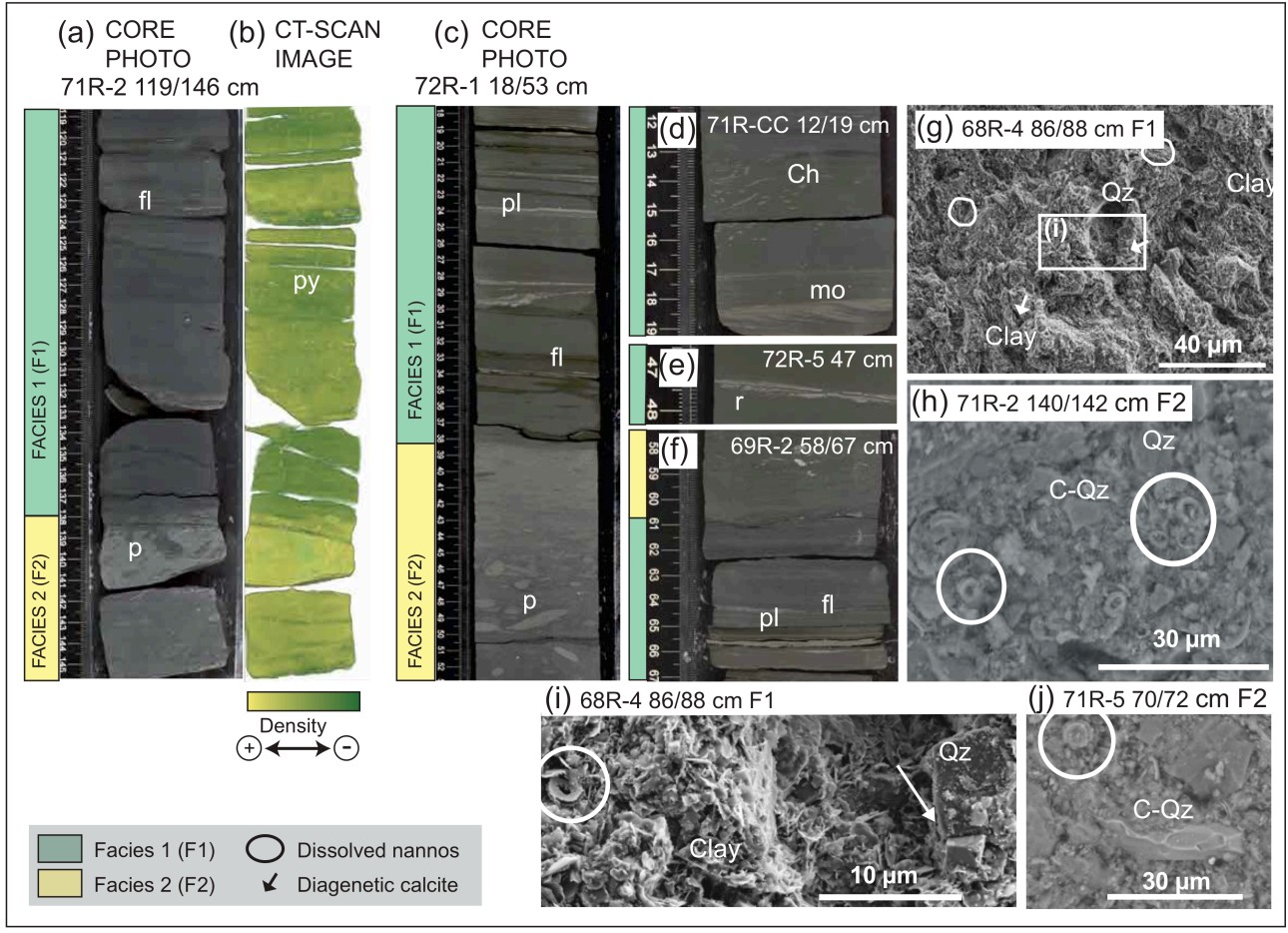

**Fig. 3:** Detailed images, CT-scans and HRSEM from Facies 1 (F1) and Facies 2 (F2). (a) Example of F1 taken from Core 71R-2 119/146 cm, showing faint laminations (fl) and bioturbation by *Planolites* (p)  (b) CT-scan 3D image of the same core interval, note the pyritized burrows (py). (c) Example of F2 taken from core 72R-1 18/53 cm). (d-f) Close-ups of laminations from F1: ripples (r), planar lamination (pl), and faint laminations (fl), with mud offshoots (mo). (d) *Chondrites* (Ch) bioturbation inside F1. (g) HRSEM image of F1 (68R-4-86/88 cm) with detritic aspect and a mudstone clay matrix, Quartz grains (Qz), diagenetic calcite (arrows), and dissolved coccoliths (circles); (h) HR-SEM image of F2 (71R-2 140/142 cm) silt sized matrix and reworked calcareous nannofossils, and conchoidal quartz grain (C-Qz); (i) Detail of dissolved coccoliths and diagenetic calcite mineral; (j) Detail of a dissolved and reworked calcareous nannofossils and a fractured conchoidal quartz (C-Qz).

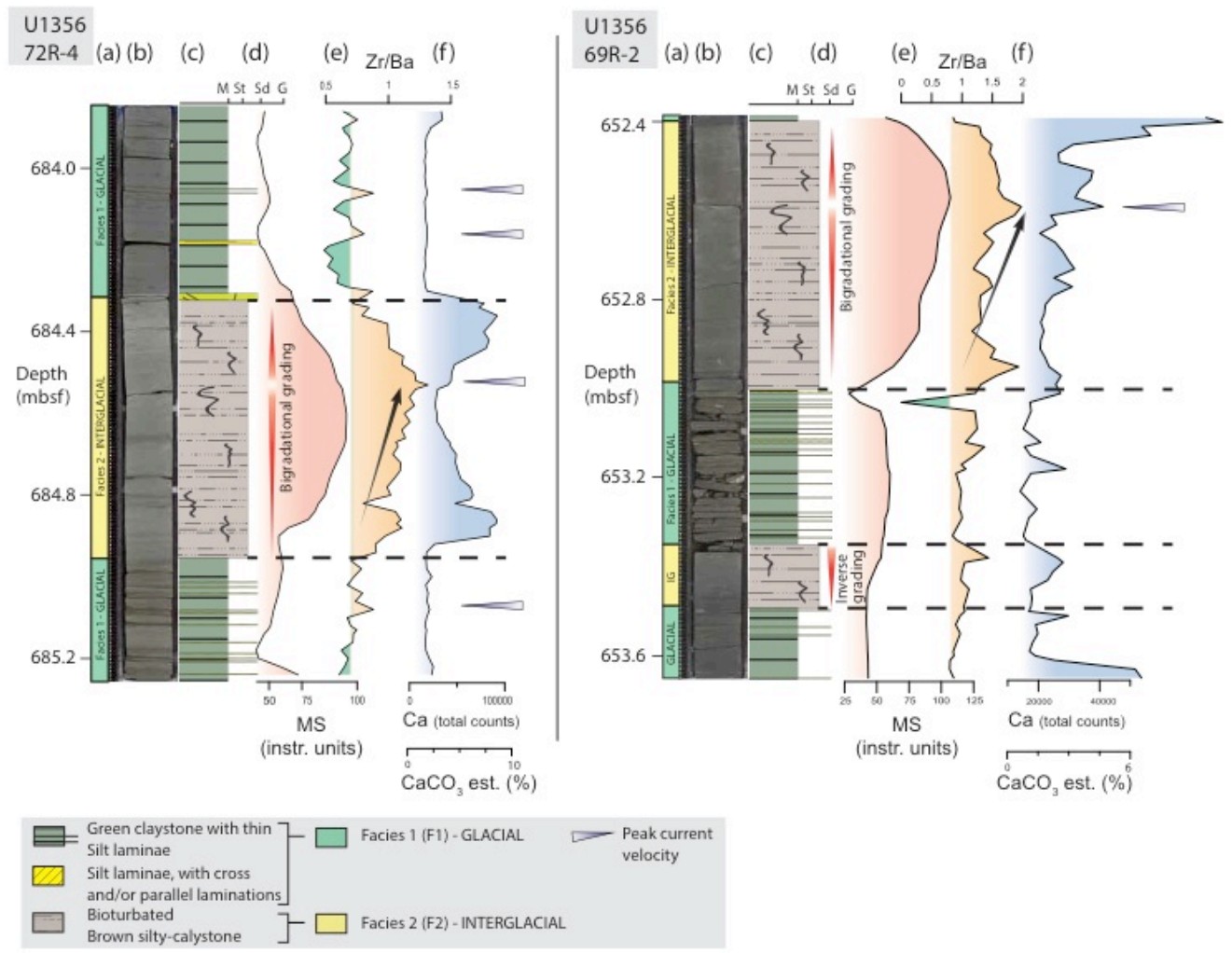

**Fig. 4:** Detailed facies characterization of two representative sections using: (a) Interpreted facies F1 and F2; a high-resolution digital image of the core sections (b), facies log (c), Magnetic susceptibility (MS) (d), XRF Zr/Ba ratio (e), and XRF calcium counts (f).

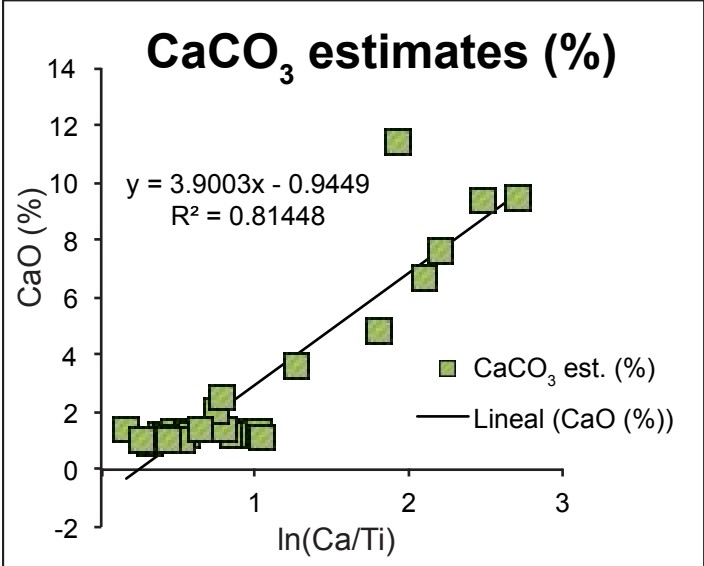

**Fig. 5:** Linear correlation between CaO% (discrete XRF) and ln(Ca/Ti) (XRF scanner) values in order to estimate carbonate contents ($CaCO_3$ est. %).

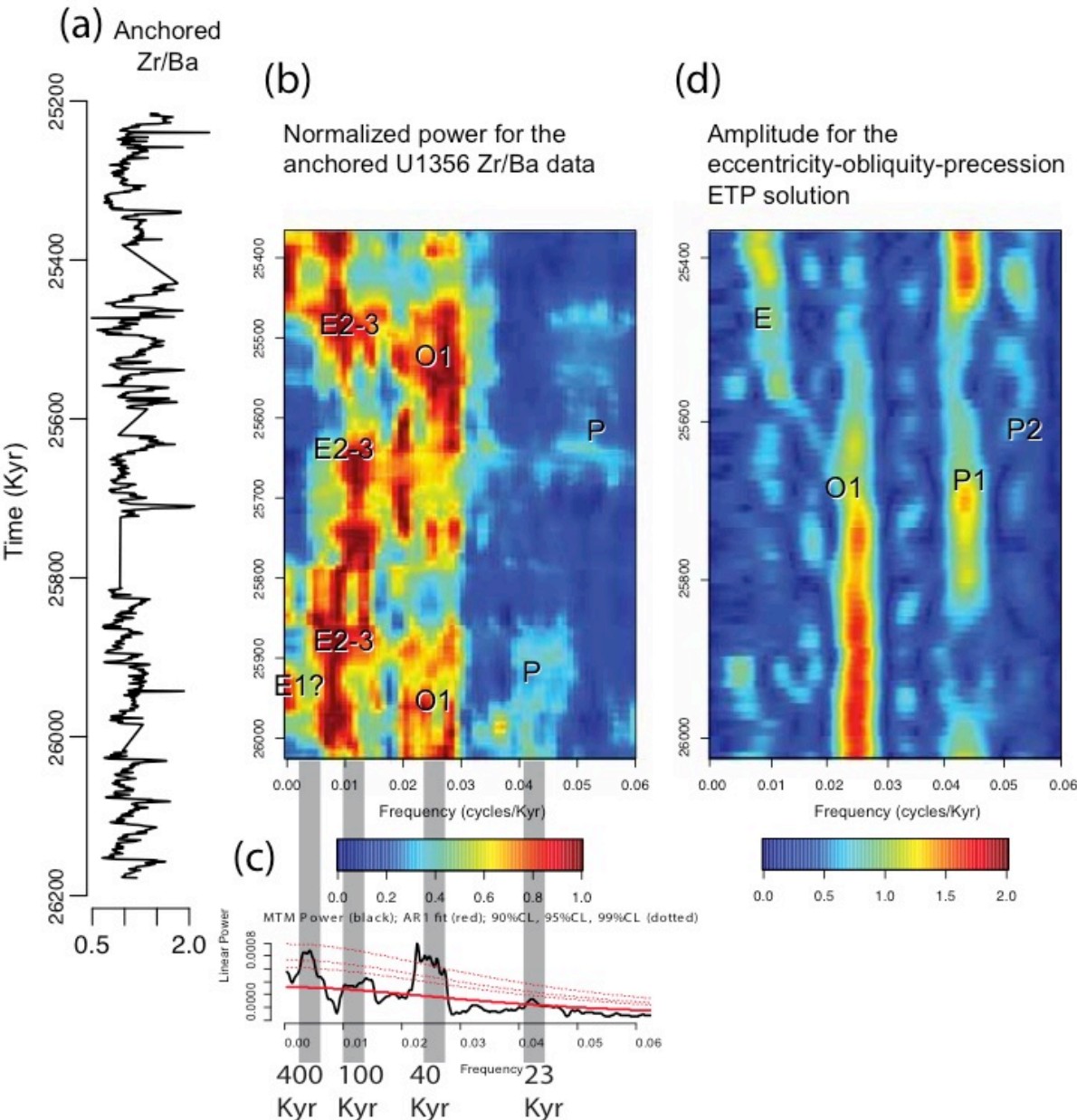

**Fig. 6:** Spectral analysis results of the Zr/Ba obliquity tuned and anchored data. (a) Zr/Ba ratio tuned with Astrochron (Meyers, 2014) and anchored to the top of the C8n.2n (o) chron. (b) EHA and (c) MTM spectral analysis on Zr/Ba tuned data. EHA normalized power with 300-kyr window with 3DPSS tapers. (d) EHA amplitude for the eccentricity-obliquity-precession ETP solution (Laskar et al., 2004) calculated for the same period of time with with 3DPSS tapers and 200-kyr window.

Paleoceanographic configuration of Wilkes Land region during the
Late Warm Oligocene (~26-25 Ma)

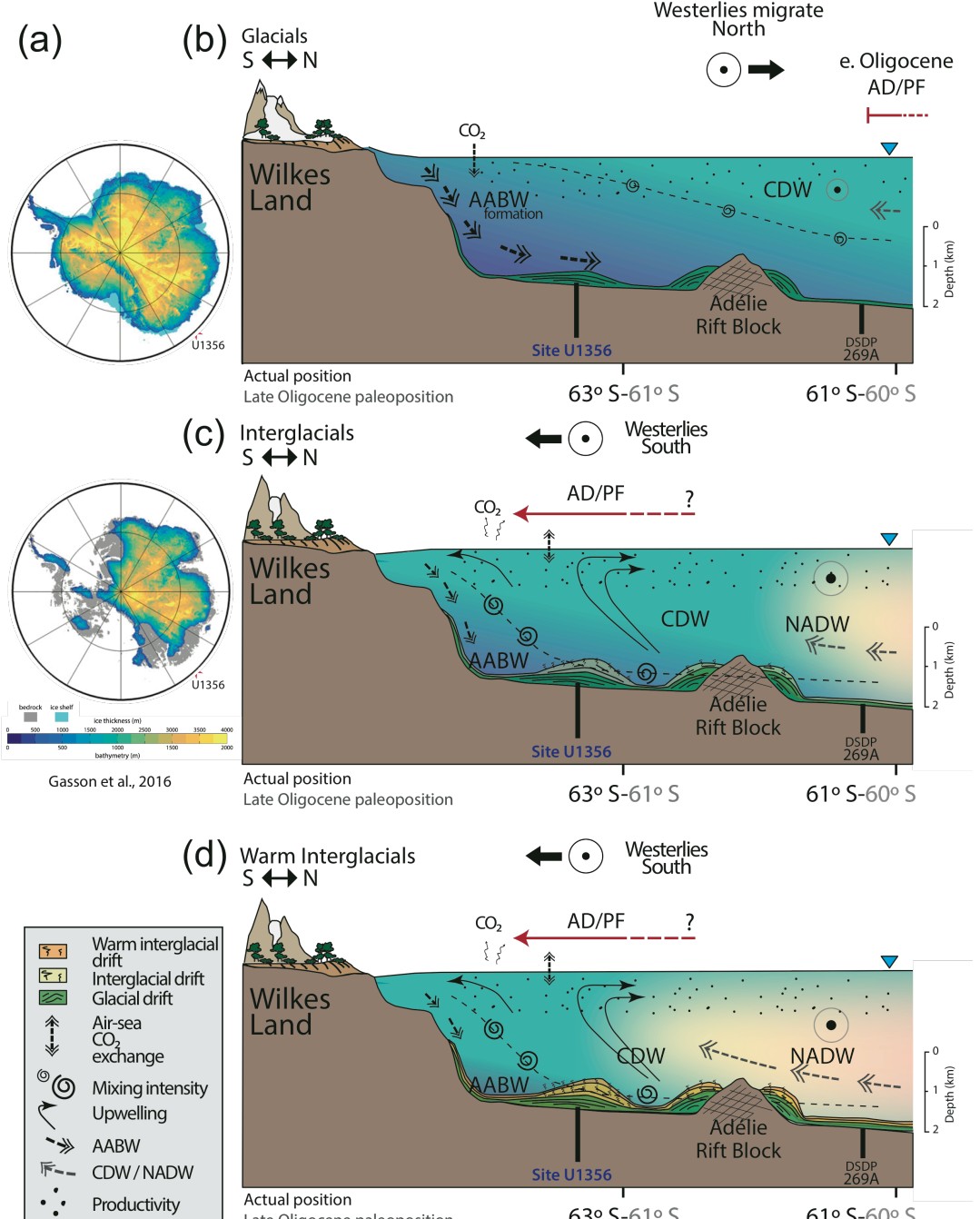

**Fig. 7:** Paleoceanographic reconstructions based on our interpretations for Facies 1 and 2. (a) Modelled ice thickness for the mid-Miocene ice sheet by Gasson et al., (2016). (b) Glacial periods with low obliquity configuration. Westerlies and Polar Front (PF) move northwards. There is enhanced proto-AABW formation. Low ventilation conditions occur at the ocean/sediment interface and mixing of waters masses is diminished. Bottom currents are weak and fluctuating, producing laminated sediments. (b) Interglacials occur during high obliquity configuration. Westerlies and the PF move southwards, close to the Site U1356. Proto-AABW formation is reduced. Intrusions of proto-CDW/NADW-like reach southernmost positions. (c) During warm Interglacials, NADW-like is enhanced and $CaCO_3$ sedimentation is more abundant. (b,c) Bottom water ventilation and upwelling are more vigorous, with stronger bottom currents that result in fully bioturbated and silty-sized sediments.