# Peer review of "Late Oligocene astronomically paced contourite sedimentation in the Wilkes Land margin of East Antarctica: insights into paleoceanographic and ice sheet configurations"

_Climate of the Past, 2017_

## Referee Comment (RC1) · Anonymous Referee #1 · 2 Jan 2018

Excellent (1)
Good (2)
Fair (3)
Poor (4)

Does the manuscript represent a substantial contribution to scientific progress within the scope of Climate of the Past (substantial new concepts, ideas, methods, or data)?

Good

Scientific quality:
Are the scientific approach and applied methods valid? Are the results discussed in an appropriate and balanced way (consideration of related work, including appropriate references)?

Good

Presentation quality:
Are the scientific results and conclusions presented in a clear, concise, and well-structured way (number and quality of figures/tables, appropriate use of English language)?

Good/Excellent

Does the paper address relevant scientific questions within the scope of CP?

Yes it does.

Does the paper present novel concepts, ideas, tools, or data?

Yes.

Are substantial conclusions reached?

Yes. Though, partially due to the nature of the data/research, many conclusions remain largely speculative.

Are the scientific methods and assumptions valid and clearly outlined?

Partially. I think that such a wide variety of data is presented, that integrating all lines of evidence is very complex. I think that the authors can improve on this point. Especially, by better outlining/introducing their approach (why each data set is presented and what it shows) and in their summary/conclusions (How the

argument (largely sedimentologal in nature) is constructed). The paleoclimatic and paleoceanographic conclusions are speculative.

Are the results sufficient to support the interpretations and conclusions?

Yes, I think so. However this research comes with large limitations of course.

Is the description of experiments and calculations sufficiently complete and precise to allow their reproduction by fellow scientists (traceability of results)?

Yes.

Do the authors give proper credit to related work and clearly indicate their own new/original contribution?

Yes.

Does the title clearly reflect the contents of the paper?

Yes. I think so. Though perhaps be more careful with the orbital interpretations. Good age control in these sediments is difficult to achieve. Perhaps replace "obliquity" with "astronomical"? Given the moderate recovery (many gaps), 1 million year length of the record, and relatively poor absolute&relative age control, I wonder if the generalization of the presumed obliquity pacing for the (entire?) Late Oligocene (as the title could suggest) is too much. Also, I wonder if contourite is the correct sedimentological description of these sediments. I realise that this argument is explored in great detail in this manuscript, however I am no sedimentologist and I wonder how these contourites compare to those from, for example, the Iberian margin. Levy et al. PNAS 2016 present 5 motives for a very proximal site. Could the lithological alterations at Wilkes Land not be linked to these motives as well? And are we still speaking of contourites then?

Does the abstract provide a concise and complete summary?

Improvements can be made. Please see below.

Is the overall presentation well structured and clear?

In general it is a very long paper with many (complex) lines of evidence. I feel that this could be outlined (signposted throughout the manuscript) a bit better. Perhaps introduce when new datasets are presented and why these data are important for this study. What questions will they help answering?

Is the language fluent and precise?

Yes.

Are mathematical formulae, symbols, abbreviations, and units correctly defined and used?

Yes.

Should any parts of the paper (text, formulae, figures, tables) be clarified, reduced, combined, or eliminated?

I think that making the manuscript more concise/focussed would help with getting the main points across.

Are the number and quality of references appropriate?

Yes. Perhaps add Levy et al. 2016 PNAS.

Is the amount and quality of supplementary material appropriate?

I have not been able to find the supplementary data online. I have not reviewed this.
* * *
Further comments:

L43: I think that the link between the data presented in this paper and ice sheet configuration is speculative at best. I would not start the abstract with such a bold claim. Delete, or move to the final line of the abstract and say something like: "we speculate on the ice sheet configurations of the Wilkes Land Basin from between 25 and 26 million years ago.

L46: Physical properties are only magnetic susceptibility. I would just say that. I would also be more precise about what geochemical techniques are presented. Key paleoceanographic/ice sheet indicators, such as fish tooth and detrital Nd are not presented. Make that clear in the abstract.

L51-54: Not a sentence. I would first present a short summary of the sedimentological result. Then say how these are interpreted. Best not to mix these up.

L58: Why lowlands? Why not topographic highs? Could your data not support both options?

L64-65: The line about spectral analysis is stuck on the end of the abstract. A strange place to present new results/interpretations. I would advice to end the abstract with the biggest (although perhaps speculative) conclusions. Not new information about the sedimentological/statistical description.

L137: Just say magnetic susceptibility of the bulk sediment.

L184: Cite individual chapters of the Gradstein volume. In this case Vandenberghe et al. (the Paleogene chapter).

L191: I have not been able to find supplementary information online. Did I miss anything?

L205: Which lab was used for this analysis?

L235: Al counts are often very sensitive to coring disturbances. I think this should be mentioned and that the authors should be careful with the interpretation of Al counts from heavily disturbed sediments.

L256: Crucial point. How was the data anchored (tuned) to obliquity? This point needs to be described and explored in much more detail. What assumptions are underlying the tuning? The readers need to know how certain the authors are about the age model/tuning etc. What is the room for improvement?

L256 and L260/261 mention two different tuning targets. One based on obliquity, the other on eccentricity, obliquity and precession. Please clarify.

L270: Please clarify how your sedimentological descriptions are better than the shipboard description. How did you improve?

L435: Perhaps compare to Levy et al?

L520: Could there be other reasons why there is no IRD at your site? (Absence of evidence is not necessarily evidence of absence)

L580: I do not understand how the authors conclude that ice was present in the lowlands. Are topographic highs not a much more likely location of land ice?

L603: do the authors mean that the palynomorphs are partially oxidized/poorly preserved? Please clarify if that is the case.

L681: What is the evidence that northern component waters were reaching this site that is located so far south in the modern and in the Oligocene? The evidence for NCW in the Oligocene needs to be better explained/this point needs to be presented/supported in a much better way.

L689: Noise and gaps in time series are two different things. Please correct.

L697: Why would precession suggest a dynamic ice sheet? Are there other mechanisms that could be thought of to explain a potential precession beat in your data?

L711: More caution needs to be taken when interpreting tuned records. Many assumptions are implicit.

L744: Nd evidence is needed before this can be suggested with any level of confidence. This is just speculation in my opinion. Please rephrase.

L749: Add "in the Wilkes Land Basin"

L753: how is this conclusion supported by the data? No ice volume estimates are presented.

Despite my (hopefully) constructive criticism, I am very supportive of this paper. I hope to see it published soon in Climate of the Past and wish to congratulate the authors on a very nice study.

---

## Referee Comment (RC2) · S. Pekar (Referee) · 26 Jan 2018

This is a review of the manuscript entitled "Evidence for Substantial Variations in Ice Volume During the middle Eocene 'doubthouse'", by A. Salabarnada and others for consideration in the journal called Climate of the Past.

This manuscript describes an exceptional sedimentary archive of Antarctic glacial history during the late Oligocene from IODP Site U1356. A great amount of data collection and thoughtful interpretations have gone into this manuscript and the authors should

be commended for this.

I will have to admit that this manuscript review slipped through the cracks and I did not get a chance to start the review until the night before the deadline. Therefore, I will only bring up the most important points that I believe the authors need to address.

My first major concern with the manuscript was it stating about the lack of IRD in their studied interval is taken to indicate the relative absence of marine-terminating ice sheets at the nearby margin. I have to differ with this important result as in another study by D. Hauptvogel, identified IRD's in the same interval at Site U1356. He did this by counting grains larger than 150 microns in many samples within this interval. In approximately 25% of his samples within the same interval used in this manuscript contained significant numbers of >150 micron grains usually between 2 and 5%. In addition the sand percent for the Late Oligocene is not much less than what is seen in the early Oligocene from Site U1356. I remember that Dr. Hauptvogel spoke with the lead author back in 2016 and he sent her his sand percentage data as well as visually showed her the work he had done on the sand fraction. While there could be an argument that bottom currents could move fine sand size grains, medium size grain sized grains were also identified. So I am not sure if grains larger than 150 microns could easily be moved from the Mertz Shear Fracture (source of the grains based on Ar Ar dating) to Site U1356 only by bottom water currents. At the very least, the authors need to discuss and explain this point far better.

I also have some concerns with the age model, as there is only one good tie point for the late Oligocene, which is at 26.1 Ma. The spectral analysis looks good in figure 6 until 25.8 Ma but looks far more uncertain above, probably because the age model is not well resolved.

I think that the statement about ice in the lowlands versus the coast or versus the highlands is a bit speculative. Especially since there are no data that estimates ice volume in this manuscript as well as that there are grains larger than 150 microns that

occur throughout the late Oligocene section at Site U1356.

The evidence of NCW to explain the glacial /interglacial changes seen here are a bit thin. The papers cited are explaining long term trends not at Milankovitch timescales. I would suggest that this be discussed in a better way.

I don't understand how precession suggests a dynamic ice sheet.

The last paragraph of the conclusions is speculative as there is little data to support it.

In conclusion, I believe that this manuscript needs to be and should be published as it has an excellent data set and has a number of new and exciting conclusions. I hope that my concerns are taken in a constructive manner in the hope of improving some of the weaker aspects of the manuscript so it can stand the test of time in the literature. Since it is obvious who I am based on my comments, I decided not to remain anonymous. Therefore, I invite the authors to contact me if they have any questions or would like to discuss any of my comments.

---

## Author Comment (AC2) · 5 Mar 2018

We apologize for the late response but I have been embarked on a research cruise in Antarctica with very limited internet connection. Firstly, we would like to thank the reviewer, Dr. Steve Pekar, for his comments and constructive suggestions, which will improve the manuscript. Below, we address the main concerns of Dr. Pekar in cursive font:

**Concern 1**: "…major concern with the manuscript was it stating about the lack of IRD in their studied interval is taken to indicate the relative absence of marine-terminating ice sheets at the nearby margin. I have to differ with this important result as in another study by D. Hauptvogel, identified IRD's in the same interval at Site U1356. He did this by counting grains larger than 150 microns in many samples within this interval. In approximately 25% of his samples within the same interval used in this manuscript contained significant numbers of >150 micron grains usually between 2 and 5%. In addition, the sand percent for the Late Oligocene is not much less than what is seen in the early Oligocene from Site U1356. I remember that Dr. Hauptvogel spoke with the lead author back in 2016 and he sent her his sand percentage data as well as visually showed her the work he had done on the sand fraction. While there could be an argument that bottom currents could move fine sand size grains, medium size grain sized grains were also identified. So I am not sure if grains larger than 150 microns could easily be moved from the Mertz Shear Fracture (source of the grains based on Ar/Ar dating) to Site U1356 only by bottom water currents. At the very least, the authors need to discuss and explain this point far better."

*We agree with the reviewer that the absence of IRD in our studied interval is not to be taken as the sole evidence for lack of marine terminating ice sheets in the Wilkes Land margin. We first want to point out that we reach this conclusion not only based on the absence of IRDs but also other supporting evidence such as: (1) the lack of sea ice indicated by the dynocists (Bijl et al companion paper to this one in CP); (2) elevated sea surface temperature (Hartmann et al. companion paper to this one in CP); and (3) palynomorph data (Salzmann et al 2016). The reviewer however, questions our conclusion because, in the study of Dr. Hauptvogel, IRD grains were identified, which contradicts our findings. As Dr. Pekar mentions, we discussed and compared our data sets during a meeting but we did not had his sand percentage data from his PhD thesis, that was available online (Hauptvogel, 2015) (https://academicworks.cuny.edu/cgi/viewcontent.cgi?referer=&httpsredir=1&article=1980&context=gc_etds). We like to note that the grains interpreted as IRD in Dr. Haupvogel´s thesis are, as Dr. Pekar indicates in his review, those that have a grain-size >150 microns. We like to emphasize that in Dr. Haupvogel thesis, these grains are considered as IRD based on the assumption that, as stated by Dr. Pekar in his review, grains larger than 150 microns cannot easily be moved from the Mertz Shear Fracture to Site U1356 only by bottom water currents. This implies that Dr. Hauptvogel assumes that sand grains >150microns can only be delivered to the*

*continental rise site U1356 by icebergs (as also stated in page 48 from Hauptvogel 2015, PhD Thesis). However, globally, sand and gravels can be transported to deep areas of the basins by multiple processes such as are Mass Transport Deposits (MTDs), turbidity currents, hyperpycnal flows, etc. In fact, during Expedition 318 moderately-to-well sorted, sandy granule-pebble sediments grading upwards into well-sorted fine, crudely stratified sands were recovered from Site U1355 at 3729 m water depth at the mouth on one of the submarine channels (Escutia et al., 2011). Also on the Wilkes Land, a sample collected by the USNS Eltanin from one of the Wilkes Land continental rise channels, has high-content in sand and rock fragments (Payne and Conolly 1972; Escutia et al., 2000). These findings point to delivery of very coarse material from the continental shelf to the continental rise by gravitational processes. Therefore, the assumption that >150 microns sand grains can only be delivered by icebergs to where our site is located is not accurate. Also, note that in our manuscript, we do not claim the sand to be delivered by bottom currents as implied in Dr. Pekar´s review. Instead, our facies analyses points to sediments delivered to where site U1356 is located on the continental rise, dominantly by gravity flows and hemipelagic sedimentation, which are then reworked by bottom currents.*

**Concern 2**: "I also have some concerns with the age model, as there is only one good tie point for the late Oligocene, which is at 26.1 Ma. The spectral analysis looks good in figure 6 until 25.8 Ma but looks far more uncertain above, probably because the age model is not well resolved".

*We use the three paleomagnetic chrons by Tauxe et al. 2012. Using the two different statistical approaches provided in the manuscript and the Suplementary, we arrived to a well-resolved age model that considers two strong tie points: one in the top of the studied core interval and another one at the bottom (Chron C8n.1n (o), 25.260 Ma, at 643.37 mbsf; and C8n.2n (o), 25.900 Ma, at 678.98 mbsf). We will clarify in the revision that the age control, although reliable is of is low-resolution.*

**Concern 3**: "I think that the statement about ice in the lowlands versus the coast or versus the highlands is a bit speculative. Especially since there are no data that estimates ice volume in this manuscript as well as that there are grains larger than 150 microns that C2 occur throughout the late Oligocene section at Site U1356.

*We agree with the reviewer that our data does not provide ice volume estimates. We see the confusion caused by the way the sentence is written. Of course, the ice caps and glaciers occupied lowlands as well topographic highs. In our sentence, we mainly wanted to emphasize the different topography of the Wilkes subglacial Basin, which in the Oligocene was not yet over-deepened. We will try to clarify this by rephrasing the sentence to "These observations, supported by elevated sea surface paleotemperatures and the absence of sea-ice, suggest that between 26 and 25 Ma open water conditions prevailed and therefore glaciers or ice caps occupied the topographic highs and lowlands of the now over-deepened Wilkes Land subglacial Basin."*

**Concern 4**: " The evidence of NCW to explain the glacial /interglacial changes seen here are a bit thin. The papers cited are explaining long term trends not at Milankovitch timescales. I would suggest that this be discussed in a better way.

*We like to clarify that we do not use NCW to explain glacial/interglacial cyclicity. The cyclicity in our record is explained by the alternation of facies, which we find are astronomically forced, at 40Kyr. Based on the unusual presence of calcareous coccolithospheres in some of the intervals in our record (at <60º S latitude), we hypothesise that during higher than normal interglacials a proto-CDW may have been influenced by warmer NCW as the Polar Front was displaced to the south. Similar interpretations are provided in other studies in sediments of Pliocene-Pleistocene age around the Southern Ocean recording the striking presence of calcareous nannofossils ( Kuhn and Diekmann, 2002; Cowan et al., 2008; Villa et al., 2012)*

**Concern 5**: " I don't understand how precession suggests a dynamic ice sheet."

*We agree with the Reviewer that precession can have different interpretations in our record. Although highly speculative, as our record captures the precession frequencies, we suggested that high latitude summer insolation during late Oligocene had an influence on the continental terrigenous fraction suggesting ice melt and rapid ice-sheet volume changes as Patterson et al., (2014) also suggested for core U1361 in Wilkes Land during the Pliocene. However, given that this interpretation does not add to any of the relevant point of the manuscript and is highly speculative, we will remove it.*

**Concern 6**: "The last paragraph of the conclusions is speculative as there is little data to support it."

*We agree with the reviewer. We reformulated the paragraph in order to be more precise and expose only the data where we are confident.*

References:

Bijl, P.K., Houben, A.J.P., Hartman, J.D., Pross, J., Salabarnada, A., Escutia, C., Sangiorgi, F., 2017. Oligocene-Miocene paleoceanography off the Wilkes Land Margin (East Antarctica) based on organic-walled dinoflagellate cysts. Clim. Past Discuss. 2017, 1–43.

Cowan, E.A., Hillenbrand, C.D., Hassler, L.E., Ake, M.T., 2008. Coarse-grained terrigenous sediment deposition on continental rise drifts: A record of Plio-Pleistocene glaciation on the Antarctic Peninsula. Palaeogeogr. Palaeoclimatol. Palaeoecol. 265, 275–291.

Escutia, C., Brinkhuis, H., Klaus, A., Scientists, I.E. 318, 2011. Expedition 318 summary, in: Proceedings of the Integrated Ocean Drilling Program, Volume 318.

Escutia, C., Eittreim, S.L., Cooper, a K., Nelson, C.H., 2000. Morphology and acoustic character of the antarctic Wilkes Land turbidite systems: Ice-sheet-sourced versus river-sourced fans. J. Sediment. Res. 70, 84–93.

Hartman, J.D., Sangiorgi, F., Salabarnada, A., Peterse, F., Houben, A.J.P., Schouten, S., Escutia, C., Bijl, P.K., 2017. Oligocene TEX86-derived seawater temperatures from offshore Wilkes Land (East Antarctica). Clim. Past Discuss. 2017, 1–31.

Hauptvogel, D.W., 2015. The State Of The Oligocene Icehouse World : Sedimentology , Provenance , And Stable Isotopes Of Marine Sediments From The Antarctic Continental Margin. PhD Dissertation. The City University Of New York.

Kuhn, G., Diekmann, B., 2002. Late Quaternary variability of ocean circulation in the southeastern South Atlantic inferred from the terrigenous sediment record of a drift deposit in the southern Cape Basin (ODP Site 1089). Palaeogeogr. Palaeoclimatol. Palaeoecol. 182, 287–303.

Patterson, M.O., McKay, R., Naish, T., Escutia, C., Jimenez-Espejo, F.J., Raymo, M.E., Meyers, S.R., Tauxe, L., Brinkhuis, H., Klaus, a., Fehr, a., Bendle, J. a. P., Bijl, P.K., Bohaty, S.M., Carr, S. a., Dunbar, R.B., Flores, J. a., Gonzalez, J.J., Hayden, T.G., Iwai, M., Katsuki, K., Kong, G.S., Nakai, M., Olney, M.P., Passchier, S., Pekar, S.F., Pross, J., Riesselman, C.R., Röhl, U., Sakai, T., Shrivastava, P.K., Stickley, C.E., Sugasaki, S., Tuo, S., van de Flierdt, T., Welsh, K., Williams, T., Yamane, M., 2014. Orbital forcing of the East Antarctic ice sheet during the Pliocene and Early Pleistocene. Nat. Geosci. 7, 841–847.

Payne, R.R., Conolly, J.R., Aabbott, W.H., 1972. Turbidite Muds within Diatom Ooze off Antarctica: Pleistocene Sediment Variation Defined by Closely Spaced Piston Cores. GSA Bull. 83, 481–486.

Salzmann, U., Strother, S., Sangiorgi, F., Bijl, P., Pross, J., Woodward, J., Escutia, C., Brinkhuis, H., 2016. Oligocene to Miocene terrestrial climate change and the demise of forests on Wilkes Land, East Antarctica, in: EGU General Assembly Conference Abstracts, EGU General Assembly Conference Abstracts. p. EPSC2016-2717.

Villa, G., Persico, D., Wise, S.W., Gadaleta, A., 2012. Calcareous nannofossil evidence for Marine Isotope Stage 31 (1Ma) in Core AND-1B, ANDRILL McMurdo Ice Shelf Project (Antarctica). Glob. Planet. Change 96–97, 75–86.

---

## Author Response (AR1)

We apologize for the late response but I have been embarked on a research cruise in Antarctica. We would like to thank anonymous Reviewer 1 for his/her comments and constructive suggestions, which will help to improve the manuscript. Below are our answers to the comments in black ink and italic.

Does the manuscript represent a substantial contribution to scientific progress within the scope of Climate of the Past (substantial new concepts, ideas, methods, or data)?
Good

Scientific quality:
Are the scientific approach and applied methods valid? Are the results discussed in an appropriate and balanced way (consideration of related work, including appropriate references)?
Good

Presentation quality:
Are the scientific results and conclusions presented in a clear, concise, and well-structured way (number and quality of figures/tables, appropriate use of English language)?
Good/Excellent

Does the paper address relevant scientific questions within the scope of CP?
Yes it does.

Does the paper present novel concepts, ideas, tools, or data?
Yes.

Are substantial conclusions reached?
Yes. Though, partially due to the nature of the data/research, many conclusions remain largely speculative.

Are the scientific methods and assumptions valid and clearly outlined?
Partially. I think that such a wide variety of data is presented, that integrating all lines of evidence is very complex. I think that the authors can improve on this point. Especially, by better outlining/introducing their approach (why each data set is presented and what it shows) and in their summary/conclusions (How the argument (largely sedimentological in nature) is constructed). The paleoclimatic and paleoceanographic conclusions are speculative.

*The high recovery of late Oligocene sediments during Expedition 318 provided an*

*unique opportunity to study the environmental conditions at this site that is close to the Antarctic margin. No single indicator provides a clear picture of these past high-CO2 world environments but the mutiproxy approach used here helps in testing out some of the environmental signals. The conclusions reached are by the nature of this study speculative since they are reached with data from a single site. However, the paleoclimatic and paleoceanographic conclusions are not so speculative as they may appear when we take into account that the paleoclimatic conditions are supported by the Sea Surface paleotemperatures reported by the companion paper submitted to Climate of the Past by Hartman et al.; and the paleoceanographic conditions by the paper by Bijl et al. It is unfortunate that the reviewers did not have access to these other two papers.*

*In the Methods and Discussion sections, we have added an introduction to the information provided by each of the proxies used for this study to better understand depositional settings in past climates at this poorly studied margin of Antarctica. We note that these proxies have been sparsely applied in the Antarctic region and thus, our subjective (rather than speculative) interpretations will of course be subject to further refinement with improved spatial coverage of this time periods in Antarctica, and future development of proxies to test the hypotheses developed in this paper. Lines 210, 237, 482, 492, 730.*

Are the results sufficient to support the interpretations and conclusions?
Yes, I think so. However this research comes with large limitations of course.

Is the description of experiments and calculations sufficiently complete and precise to allow their reproduction by fellow scientists (traceability of results)?
Yes.

Do the authors give proper credit to related work and clearly indicate their own new/original contribution?
Yes.

Does the title clearly reflect the contents of the paper?
Yes. I think so. Though perhaps be more careful with the orbital interpretations. Good age control in these sediments is difficult to achieve. Perhaps replace "obliquity" with "astronomical"? Given the moderate recovery (many gaps), 1 million year length of the record, and relatively poor absolute&relative age control, I wonder if the generalization of the presumed obliquity pacing for the (entire?) Late Oligocene (as the title could suggest) is too much.

*We concur with the comment by the reviewer and will substitute obliquity with astronomical in title. We also changed "implications" with "insights".*
*Title has been changed.*

Also, I wonder if contourite is the correct sedimentological description of these sediments. I realise that this argument is explored in great detail in this manuscript, however I am no sedimentologist and I wonder how these contourites compare to those from, for example, the Iberian margin. Levy et al. PNAS 2016 present 5 motives for a very proximal site. Could the lithological alterations at Wilkes Land not be linked to these motives as well? And are we still speaking of contourites then?

*We appreciate the candid comment of the Reviewer indicating that he is not a sedimentologist. Contourites in any setting and location refer to sediments deposited or significantly affected by the action of bottom currents, despite their origin. In the Wilkes Land Site U1356, the sediments deposited during glacial and interglacial cycles, which are dominantly gravity flows and hemipelagites, respectively, are reworked by bottom currents resulting in the sediments recovered at this site. Contourites from the Iberian margin are also the result of reworking of downslope and hemipelagic sedimentation. Contrary to turbidite deposits, contourites do not exhibit a "type contourite facies association model or motif" but contourite facies/structures (i.e., laminated vs bioturbated, etc) are common to all bottom current deposits (see for example the review paper by Rebesco et al., 2014). Levy et al., PNAS 2016, shows a stacking patterns of different motifs recovered from the McMurdo Sound coastal sector of the Ross Sea by the ANDRILL2A. Levy et al., interpret the sedimentary cycles represented by the motifs in terms of advances and retreats of the ice sheet grounding line forced by eccentricity. Therefore, the motifs in the Levy et al paper result from sedimentary processes associated with the **direct influence of grounding line advances and retreats in a coastal setting**. Our record is a **distal marine record**. Therefore, our site receives sediment input from the continent (which provides an indirect record for continental glaciation) and the rain of hemipelagic materials that are then reworked by ocean currents. In both Levy's et al. paper and ours, we interpret the alternation in motifs and facies to be astronomically forced.*

Does the abstract provide a concise and complete summary?
Improvements can be made. Please see below.
*Abstract has been modified in order to make it more clear, concise and shorter.*

Is the overall presentation well structured and clear?
In general it is a very long paper with many (complex) lines of evidence. I feel that this could be outlined (signposted throughout the manuscript) a bit better. Perhaps introduce when new datasets are presented and why these data are important for this study. What questions will they help answering?

*We understand the multiproxy approach used in this study can make it hard to follow the different lines of evidence. At present, each of the indicators used for this study and their relevance is explained in the Material and Methods section. However, to address this concern of Reviewer 1, we will introduce a brief outline of the relevance of the indicators used in each of the subsections in the Results.*
*Combined with earlier comment, we tried to clarify and better structure our text.*
*Changes have been made all through the text.*
*Lines 210, 237, 482, 492, 730.*

Is the language fluent and precise?
Yes.

Are mathematical formulae, symbols, abbreviations, and units correctly defined and used?
Yes.

Should any parts of the paper (text, formulae, figures, tables) be clarified, reduced,

combined, or eliminated?
I think that making the manuscript more concise/focussed would help with getting the main points across.

*We will work to make the revised version of this manuscript more concise.*
*We hope the revised version of this manuscript is now more concise.*

Are the number and quality of references appropriate?
Yes. Perhaps add Levy et al. 2016 PNAS.

*In our paper, we have established comparisons with the environmental setting between the Wilkes Land and the Ross Sea. We have focussed on coeval records to those we are studying, both coastal (CPR, Barrett, 2007) and distal (DSDP Site 270, Kemp and Barrett, 1975) sites. We will introduce the reference to Levy et al 2016 in the "4.2 Ice sheet configuration during the warm late Oligocene" chapter, in line 553, by adding  "Also, a dynamic ice sheet is described for the early Miocene coastal section of AND-2A with glacial and interglacial advances and retreats of the EAIS (Levy et al., 2016), that could have a similar paleotopographic configuration to that for the Oligocene."*

*Reference has been introduced in Line 675.*

Is the amount and quality of supplementary material appropriate?
I have not been able to find the supplementary data online. I have not reviewed this.

*It is unfortunate that if appears that the Reviewer did not have access to the Supplementary material when these were available online as they were submitted with the manuscript.*

Further comments:

L43: I think that the link between the data presented in this paper and ice sheet configuration is speculative at best. I would not start the abstract with such a bold claim. Delete, or move to the final line of the abstract and say something like: "we speculate on the ice sheet configurations of the Wilkes Land Basin from between 25 and 26 million years ago.

*We will proceed in the revision as advised by the Reviewer. However, our claim on the retreated ice sheet is reinforced by several lines of evidence: (1) How the late Oligocene interval studies compares to the rest of the Oligocene and early Miocene sediments (presented in the supplementary materials to which the Reviewer unfortunately did not have access). Earlier Oligocene and Miocene sections contain Ice Rafted Debris, suggesting an extended ice sheet. No IRD was found in the studied interval and we argue this could be indicative of less extensive ice sheets. (2) As referred in the paper and more extensively covered in the companion paper to this one by Bijl et al., dynocists indicate that during the studied interval there is no evidence for sea ice suggesting a warmer setting and reduced ice sheets during the studied late Oligocene interval. Sea ice species are however present in other Oligocene and Miocene intervals from U1356 core. (3) High Sea Surface Temperature reconstructions as shown in the companion paper to this one by Hartmann et al.  that*

support the sea ice free scenario. (4) Reconstructions derived from fossil pollen in Site U1356 suggesting high terrestrial temperatures (Salzmann et al., 2016; Strother et al, in prep).

*We have made changes to clarify that our interpretations are based in several lines of evidence. All geological studies are interpretative, with various degrees of uncertainties. Speculative however implies we do not have clear evidence, which is not the case as outlined above in our rebuttal.*
*Line 40.*

L46: Physical properties are only magnetic susceptibility. I would just say that. I would also be more precise about what geochemical techniques are presented. Key paleoceanographic/ice sheet indicators, such as fish tooth and detrital Nd are not presented. Make that clear in the abstract.

*We will follow the suggestion by the Reviewer. For the physical properties however, in addition to the magnetic susceptibility, we also use density.*
*L. 45*

L51-54: Not a sentence. I would first present a short summary of the sedimentological result. Then say how these are interpreted. Best not to mix these up.

*We will follow the advise of the Reviewer*
*L. 52*

L58: Why lowlands? Why not topographic highs? Could your data not support both options?

*We see the confusion caused by the way the sentence is written. Of course the ice caps and glaciers occupied as well topographic highs. We wanted to mainly emphasize the different topography of the Wilkes subglacial Basin compared to today, which in the Oligocene was not yet over-deeepened. We will try to clarify this by rephrasing the sentence to indicate "These observations, supported by elevated sea surface paleotemperatures and the absence of sea-ice, suggest that between 26 and 25 Ma open water conditions prevailed and therefore glaciers or ice caps occupied the topographic highs and lowlands of the now over-deepened Wilkes Land subglacial Basin."*

*L. 62 changed as follows: "…these evidences suggest that glaciers or ice caps likely occupied the topographic highs and lowlands of the now marine Wilkes Subglacial Basin (WSB)."*

L64-65: The line about spectral analysis is stuck on the end of the abstract. A strange place to present new results/interpretations. I would advice to end the abstract with the biggest (although perhaps speculative) conclusions. Not new information about the sedimentological/statistical description.

*We will rewrite following the advise of the Reviewer.*
*Moved at Line 55.*

L137: Just say magnetic susceptibility of the bulk sediment.

*We will rewrite as advised.*
*Line 150*

L184: Cite individual chapters of the Gradstein volume. In this case Vandenberghe et al. (the Paleogene chapter).

*We will cite as advised.*
*Line 205*

L191: I have not been able to find supplementary information online. Did I miss anything?

*It is unfortunate that the supplementary materials were not found since they are online and were submitted at the same time as the rest of the manuscript. The Supplementary information provides more detail regarding the spectral analysis applied to our datasets and also explains in more detail the sedimentary section from the early Oligocene to the early Miocene.*
*First appearance Line 216 inside Facies Analysis section*

L205: Which lab was used for this analysis?

*CT-scans were done at the Kochi Core Center (KCC) lab (Japan). It was stated in the text (L201) but we clarified it.*
*Line 227*

L235: Al counts are often very sensitive to coring disturbances. I think this should be mentioned and that the authors should be careful with the interpretation of Al counts from heavily disturbed sediments.

*We agree with the reviewer. Although we don't have core disturbances all along our studied section, we detected that Al and Si elements collected by the continuous X-Ray Fluorescence (XRF) scanner present more than one order of magnitude gains although the sediments were not deformed, and therefore they were not used. To overcome this problem, XRF analyses in discrete samples from non-deformed intervals were also conducted and are the ones used in our research. We clarified in the text that the interval for which we collected XRF data did not show core distrubances (L233).*
*Line 258. We added, "Data points from disturbed intervals in the core face (i.e., slight fractures and cracks) were removed."*

L256: Crucial point. How was the data anchored (tuned) to obliquity? This point needs to be described and explored in much more detail. What assumptions are underlying the tuning? The readers need to know how certain the authors are about the age model/tuning etc. What is the room for improvement?

*The information requested by the reviewer is contained in the supplementary materials to which, unfortunately, the reviewer did not get access. To avoid further confusions, we also will add a sentence in the main text of the manuscript to provide*

*information about anchoring the time series. For the main research we used the Evolutive Average Spectral Misfit method (Meyers, 2014) for the astrochronologic testing, that was evaluated using ETP (eccentricity, obliquity and precession) target from La04 (Laskar et al., 2004). Afterwards, an astronomical tuning is done by using the Frequency domain minimal tuning (Meyers et al., 2014) where spatial frequencies are afterwards converted to sedimentation rates using the average period of 41 Kyr/obliquity. Time series is afterwards anchored to our paleomagnetic tie points. We added a sentence in methods section and also in the result section.*
*In the supplementary data, there is also another tuning done for initial evaluation of the time series, where we tested with Analyseries method (Paillard et al., 1996) by filtering our data in depth scale and comparing it to the Obliquity solution of La04 (Laskar et al., 2004).*

*Clarification added in Line 293 in methods section and in Line 468 in results section.*

L256 and L260/261 mention two different tuning targets. One based on obliquity, the other on eccentricity, obliquity and precession. Please clarify.

*Related and answered with the previous comment.*

L270: Please clarify how your sedimentological descriptions are better than the shipboard description. How did you improve?

*Shipboard, sedimentologists describe the sections as cores are opened. Shipboard descriptions, although thorough, are preliminary since there is no time to look at the cores in the detail and the context is often lost because of changing work shifts and describers. Shipboard descriptions interpreted deposition during the studies interval to be dominated by hemipelagic and turbidity flows/bottom current processes. Post-cruise, we had a chance to re-describe all core sections in detail and by the same group of people, which included experts in turbidite and contourite deposits (not always easy to differentiate). This resulted in the very detailed lithological log presented in this paper. In addition, the integration of the detailed lithological log with magnetic susceptibility (collected shipboard), continuous/discrete XRF data, high-resolution images and CT-Scans and SEM images (obtained in the frame of this study), allowed us to further characterise the facies.*

*Line 215, 307, we clarified that the new lithologic log was constructed by re-describing the cores at mm to cm-scale resolution in comparison with the low-resolution descriptions conducted shipboard.*

L435: Perhaps compare to Levy et al?

*In this part of the discussion we focus our comparisons to facies from different settings that are similar to site U1356, mainly around East Antarctic margin. AND-2 from Levy et al., obtained sediments from a coastal site.*
*We cited Levy et al., 2016 in Line 675.*

L520: Could there be other reasons why there is no IRD at your site? (Absence of evidence is not necessarily evidence of absence)

*We agree with the reviewer that the absence of IRD cannot be directly linked to a*

*retreated ice sheet. However, as mentioned earlier, our interpretations regarding the lack of an extended ice-sheet similar to the one existing in the earlier Oligocene is not only based on the absence of IRD's but in several lines of evidence, as mentioned before, which include: (1) How the late Oligocene interval studies compares to the rest of the Oligocene and early Miocene sections (presented in the supplementary materials to which the Reviewer unfortunately did not have access). Earlier Oligocene and Miocene sections contain Ice Rafted Debris, suggesting an extended ice sheet. No IRD was found in the studied interval and we argue this could be indicative of less extensive ice sheets. (2) No evidence for sea ice indicated by dynocists and reported in detail in the companion paper to this one by Bijl et al., suggesting a warmer setting and reduced ice sheets. Sea ice species are however present in other Oligocene and Miocene intervals. (3) High Sea Surface Temperature reconstructions reported in the companion paper to this one by Hartmann et al that support the sea ice free scenario. (4) Reconstructions derived from fossil pollen in Site U1356 suggesting high terrestrial temperatures (Salzmann et al., 2016; Strother et al, in prep). In addition, we compare the environmental setting during the studied late Oligocene interval to iceberg modelling studies conducted in Pliocene sediments from the Wilkes Land margin by Cook et al (2014). The modelling shows that despite the high sea surface temperatures during warmer climate periods of the Pliocene, iceberg armadas were able to travel as far as to the continental rise sites in this margin.*

*We restructured the "Ice sheet configuration during the warm late Oligocene" subsection (Line 589-675) in order to clarify that the interpretations reached are not only based on the absence of evidence of IRDs but is also supported by other lines of evidence as outlined in the rebuttal.*

L580: I do not understand how the authors conclude that ice was present in the lowlands. Are topographic highs not a much more likely location of land ice?

*We agree with the Reviewer. ice sheets and/or glaciers would occupy both high- and lowlands. This agrees with the pollen assemblages in sediments from this interval (Ulrich Salzmann personal communication). Although what we wanted to note is that ice would be occupying the non-overdeepened Wilkes subglacial basin. We will rephrase this in the revised version to make sure it is clear.*

*Line 655. We rephrased the paragraph. We changed it in the abstract and conclusions too.*
*Line 63, 625, 869.*

L603: do the authors mean that the palynomorphs are partially oxidized/poorly preserved? Please clarify if that is the case.

*We will address the text in order to make clear that Palynomorphs have good preservation, and that don't show notable changes in their preservation between F1 and F2 (companion paper to this by Bijl et al.,).*

*Included "good preservation" in Line 699.*

L681: What is the evidence that northern component waters were reaching this site that is located so far south in the modern and in the Oligocene? The evidence for NCW in the Oligocene needs to be better explained/this point needs to be presented/supported in

a much better way.

*We will improve our discussion regarding this point in the text. We consider that as Circumpolar Deep Water is a mixture of AABW and also the NADW and the northern component waters (NCW), we interpret that during warmer times, and also due to the influence of the shifted Polar Fronts to the South during interglacials, NCW would have a higher influence on the proto-CDW, and thus, shifting the chemical characteristics towards a carbonate friendly environment. The presence of preserved coccoliths in such southernmost positions in Antarctica, and in the continental rise is rare, as many studies correlate coccoliths with the presence of a carbonated and warmer water mass shifting south, being the NCW or the NADW in the actual configuration of the ocean.*

*We rephrased our manuscript in order to make clearer the argument as follows: "Circumpolar Deep Water (CDW) is a mixing of abyssal, deep, and intermediate water masses, that includes AABW and NADW nowadays (Johnson, 2008). During warmer interglacials, the influence of more northern-sourced water masses into the proto-CDW, relative to Antarctic-sourced, could enable carbonate productivity as seen in the interglacial facies with coccolitosphere remains (Fig. 7c)."*
*We restructured the whole paragraph of the discussion and added some references to clarify our interpretations about the influence of North Component Waters at the site.*
*Line 730-808.*

L689: Noise and gaps in time series are two different things. Please correct.

*This will be corrected on the revised version of the manuscript.*
*We added "discontinuous due to gaps" in Line 828.*

L697: Why would precession suggest a dynamic ice sheet? Are there other mechanisms that could be thought of to explain a potential precession beat in your data?

*We agree with the Reviewer that precession can have different interpretations in our record. Although highly speculative, as our record captures the precession frequencies, we suggested that high latitude summer insolation during late Oligocene had an influence on the continental terrigenous fraction suggesting ice melt and rapid ice-sheet volume changes as Patterson et al., (2014) also suggested for core U1361 in Wilkes Land during the Pliocene. However, given that this interpretation does not add to any of the relevant point of the manuscript and is highly speculative, we will remove it.*
*This was stated in addition to Obliquity. We have reworded to highlight it is responded dynamically to orbital forcing at a range of frequencies.*
*We rephrased.*
*Line 835: "In addition to obliquity precession is also present, which implies a dynamic response of the EAIS and offshore oceanic water masses to orbital forcing."*

L711: More caution needs to be taken when interpreting tuned records. Many assumptions are implicit.

*We agree with the reviewer, and in no way are our results dependent on the tuning*

*applied. Line 711 refers to a previous study by Palike et al., which highlight an obliquity "heartbeat" to the climate system in the Oligocene from deep sea sediments in the Pacific. We support this observation with evidence directly from the Antarctic margin, and in no way is this dependent on tuning. We use distinct tie points and extensive statistical tests to identify the orbital periods (sup info – ASM method)*

L744: Nd evidence is needed before this can be suggested with any level of confidence. This is just speculation in my opinion. Please rephrase.

*We agree with the reviewer that Nd isotopes are a good evidence of distinct water masses. However, no fish teeth were recovered from this interval to conduct these studies. The chemistry of the water mass influences the elemental concentrations and also can give paleoceanographic information. For example, bottom waters chemistry affects the preservation of carbonates in sediments. Here, we postulate that the presence of nannofossils in site U1356 is enhanced due to more carbonated and warmer waters, less corrosive to carbonate, as are the warmer north component waters (NADW-like), that are entrained and mixed within the circumpolar deep waters (proto-CDW), that bath the basins of Antarctica.*

L749: Add "in the Wilkes Land Basin"

*Corrected.*
*Line 897.*

L753: how is this conclusion supported by the data? No ice volume estimates are presented.

*We agree with the reviewer. We left that there is a retreat of the ice sheet in the Wilkes Land Basin but we took out the processes that control the melting of the ice sheet.*
*We took out the conclusion. And we added a part of the paragraph to the discussions section in Line 806.*

Despite my (hopefully) constructive criticism, I am very supportive of this paper. I hope to see it published soon in Climate of the Past and wish to congratulate the authors on a very nice study.

References:

[revised manuscript text omitted]

Corrections and additional comments made to the rebuttals submitted to Climate of the Past in March 2018 are indicated with red ink and italic.

Answer to Referee Dr. S. Pekar for the interactive comment on "Late Oligocene obliquity-paced contourite sedimentation in the Wilkes Land margin of East Antarctica: implications for paleoceanographic and ice sheet configurations" by A. Salabarnada et al.

We apologize for the late response but I have been embarked on a research cruise in Antarctica with very limited internet connection. Firstly, we would like to thank the reviewer, Dr. Steve Pekar, for his comments and constructive suggestions, which will improve the manuscript. Below, we address the main concerns of Dr. Pekar in cursive font:

**Concern 1**: "…major concern with the manuscript was it stating about the lack of IRD in their studied interval is taken to indicate the relative absence of marine-terminating ice sheets at the nearby margin. I have to differ with this important result as in another study by D. Hauptvogel, identified IRD's in the same interval at Site U1356. He did this by counting grains larger than 150 microns in many samples within this interval. In approximately 25% of his samples within the same interval used in this manuscript contained significant numbers of >150 micron grains usually between 2 and 5%. In addition, the sand percent for the Late Oligocene is not much less than what is seen in the early Oligocene from Site U1356. I remember that Dr. Hauptvogel spoke with the lead author back in 2016 and he sent her his sand percentage data as well as visually showed her the work he had done on the sand fraction. While there could be an argument that bottom currents could move fine sand size grains, medium size grain sized grains were also identified. So I am not sure if grains larger than 150 microns could easily be moved from the Mertz Shear Fracture (source of the grains based on Ar/Ar dating) to Site U1356 only by bottom water currents. At the very least, the authors need to discuss and explain this point far better."

*We agree with the reviewer that the absence of IRD in our studied interval is not to be taken as the sole evidence for (the relative) lack of marine terminating ice sheets in the Wilkes Land margin. We first want to point out that we reach this conclusion not only based on the relative absence of IRDs but also other supporting evidence such as: (1) the lack of sea ice indicated by the dynocists (Bijl et al companion paper to this one in CP); (2) elevated sea surface temperature (Hartmann et al. companion paper to this one in CP); and (3) palynomorph data (Salzmann et al 2016).*

*We like to note that the grains interpreted as IRD in Dr. Haupvogel´s thesis are, as Dr. Pekar indicates in his review, those that have a grain-size >150 microns. This is no a commonly accepted cut-off for IRD, which is 250 microns (at a minimum; See Patterson et al., and references therein) – and thus the high percentage cited are skewed by this very fine grained cut-off values. Hauptovogel do not provide detailed grain size frequency distribution to prove that these >150 microns are outliers within the grain size population. Although is possible there is some background IRD in our*

*record, we argue it is minimal compared to elsewhere in the core – and thus represents a likely minima state in ice sheet extent relative to the periods before and after. We also note that in Dr. Haupvogel thesis, assumes that sand grains >150microns can only be delivered to the continental rise site U1356 by icebergs (as also stated in page 48 from Hauptvogel 2015, PhD Thesis). However, globally, sand and gravels can be transported to deep areas of the basins by multiple processes such as are Mass Transport Deposits (MTDs), turbidity currents, hyperpycnal flows, etc. MTDs are clearly present in the U1356 core (455 to 575 mbsf), directly contradicting this assumption. In addition, moderately-to-well sorted, sandy granule-pebble sediments grading upwards into well-sorted fine, crudely stratified sands were recovered from Site U1355 at 3729 m water depth at the mouth on one of the submarine channels (Escutia et al., 2011). Also on the Wilkes Land, a sample collected by the USNS Eltanin from one of the Wilkes Land continental rise channels, has high-content in sand and rock fragments (Payne and Conolly 1972; Escutia et al., 2000). These findings point to delivery of very coarse material from the continental shelf to the continental rise by gravitational processes.*

*Also, note that in our manuscript, we do not claim the sand to be delivered by bottom currents as implied in Dr. Pekar´s review. Instead, our facies analyses points to sediments delivered to where site U1356 is located on the continental rise, dominantly by gravity flows (bringing coarse material) and hemipelagic sedimentation, which are then reworked by bottom currents. However, we concede there may be some background IRD, but it is minimal relative to other parts of the core.*

*In the Sedimentary Facies subsection of the Results, we provide evidences backed with references of coarse-grained sediments being delivered to Site U1356 and other sites on the Wilkes margin by gravity flow processes (Mass Transport Deposits and turbidity currents). We have also included a paragraph in the Site Description subsection of the Methods regarding the depositional setting of Site U1356 during the late Oligocene on an incipient levee, which received overbank sediments from the turbidity flows traveling through the adjacent submarine channel. In addition, we have noted in Line 360 that "Although maybe some background IRD is present, it is minimal relative to other parts of the core".*
*Changes are found in the following lines:*
*Line 175, 350-361, 539, 589-609.*

**Concern 2**: "I also have some concerns with the age model, as there is only one good tie point for the late Oligocene, which is at 26.1 Ma. The spectral analysis looks good in figure 6 until 25.8 Ma but looks far more uncertain above, probably because the age model is not well resolved".

*We use the three paleomagnetic chrons by Tauxe et al. 2012. Using the two different statistical approaches provided in the manuscript and the Suplementary, we arrived to a well-resolved age model that considers two strong tie points: one in the top of the studied core interval and another one at the bottom (Chron C8n.1n (o), 25.260 Ma, at 643.37 mbsf; and C8n.2n (o), 25.900 Ma, at 678.98 mbsf). We will clarify in the revision that the age control, although reliable is of is low-resolution.*
*We make it clear at Line 468.*

**Concern 3**: "I think that the statement about ice in the lowlands versus the coast or versus the highlands is a bit speculative. Especially since there are no data that estimates ice volume in this manuscript as well as that there are grains larger than 150 microns that C2 occur throughout the late Oligocene section at Site U1356.

*Our data can not provide ice volume estimates. We see the confusion caused by the way the sentence is written. Of course, the ice caps and glaciers occupied lowlands as well topographic highs. In our sentence, we mainly wanted to emphasize the different topography of the Wilkes subglacial Basin, which in the Oligocene was not yet over-deepened. We will try to clarify this by rephrasing the sentence to "These observations, supported by elevated sea surface paleotemperatures and the absence of sea-ice, suggest that between 26 and 25 Ma open water conditions prevailed and therefore glaciers or ice caps occupied the topographic highs and lowlands of the now over-deepened Wilkes Land subglacial Basin."*
*We rephrased the statements all around the manuscript as specified.*
*Line 63, 625, 869.*

**Concern 4**: " The evidence of NCW to explain the glacial /interglacial changes seen here are a bit thin. The papers cited are explaining long term trends not at Milankovitch timescales. I would suggest that this be discussed in a better way.

*We like to clarify that we do not use NCW to explain glacial/interglacial cyclicity. The cyclicity in our record is explained by the alternation of facies, which we find are astronomically forced, at 40Kyr. Based on the unusual presence of calcareous coccolithospheres in some of the intervals in our record (at <60º S latitude), we hypothesise that during higher than normal interglacials a proto-CDW may have been influenced by warmer NCW as the Polar Front was displaced to the south. Similar interpretations are provided in other studies in sediments of Pliocene-Pleistocene age around the Southern Ocean recording the striking presence of calcareous nannofossils ( Kuhn and Diekmann, 2002; Cowan et al., 2008; Villa et al., 2012)*
*We restructured the whole paragraph of the discussion and make it clear towards our working hypothesis about the North Component Waters adding more details and evidences.*
*Line 730-808*

**Concern 5**: " I don't understand how precession suggests a dynamic ice sheet."

*This was stated in addition to Obliquity. We have reworded to highlight it is responded dynamically to orbital forcing at a range of frequencies.*
*We rephrased.*
*Line 835: "In addition to obliquity precession is also present, which implies a dynamic response of the EAIS and offshore oceanic water masses to orbital forcing."*

**Concern 6**: "The last paragraph of the conclusions is speculative as there is little data to support it."

*We agree with the reviewer. We reformulated the paragraph in order to be more precise and expose only the data where we are confident.*
*Line 900.*


~~During interglacials, our records point to more oxygenated and ventilated conditions suggesting enhanced mixing of the water masses (Fig. 7b-c). We postulate that during interglacials westerly winds and the Polar Front are shifted south and become more aligned. Under these conditions, upwelling of deep waters is promoted, facilitating the mixing and oxygenation of surface waters that form the precursor to bottom water. Such a process would also generate increased geostrophic current velocities~~

The intervals of micritic limestone within F2 have calcareous nannofossils preserved (Fig 3d). The productivity of calcareous nannofossils and the later preservation of these coccoliths in the sediment indicate specific geochemical conditions enabling carbonate deposition and preservation. Although today nannoplankton is abundant in surface waters at the Antarctic Divergence (Eynaud et al., 1999), these rarely deposit on the deep ocean floor because of corrosive bottom waters, which dissolve calcareous rain. A number of studies in other areas of the Antarctic margin and the Southern Ocean have correlated the presence of calcareous nannofossils with the presence of temperate north component water masses (North Atlantic Deep Water-like, NADW) that intrude close to the Antarctic continent and influence the Southern Ocean during the late Oligocene (e.g., Nelson and Cooke, 2001; Pekar et al., 2006; Villa and Persico, 2006; Scher and Martin, 2008), the Miocene (DeCesare et al., 2013; Sangiorgi et al., 2018), and during more recent times such as the Quaternary (Diekman, 2007; Kemp et al., 2010; Villa et al., 2012).

The more oxygenated and ventilated conditions in our records suggest enhanced mixing of the water masses (Fig. 7b-c). We postulate that during interglacials westerly winds and the Polar Front are shifted south and become more aligned. Under these conditions, upwelling of deep waters is likely promoted, facilitating the mixing and oxygenation of surface waters that form the precursor to bottom water. Similar process has been reported for the Holocene by Peck et al. (2015). Such a process would also generate increased geostrophic current velocities of the bottom water mass, supported by the coarser grain size and heavy mineral concentrations in the bioturbated F2 facies.

Similar to what is occurring under the present warming, bottom water formation during interglacials is likely fresher and less dense due to enhanced freshwater runoff from surface and subglacial melt of the continental ice sheet (Wijk and Rintoul, 2104). Today, a reduction in the volume of the AABW is compensated by the expansion of the Circumpolar Deep Water (CDW) (Wijk and Rintoul, 2014), which forms by mixing of abyssal, deep, and intermediate water masses, including the AABW and the NADW (Johnson, 2008). ~~During interglacials, bottom water formation is likely warmer and less saline due to enhanced freshwater runoff from surface and subglacial melt of the continental ice sheet. This may allow this less dense water mass to occupy shallower depths in abyssal to intermediate ocean, and promote more vigorous mixing with oxygenated CDW (Fig. 7b). Circumpolar Deep Water (CDW) is a mixing of abyssal, deep, and intermediate water masses, that includes AABW and NADW nowadays (Johnson, 2008).dD-.as seen in the interglacial facies with~~ coccolitosphere

remains, seen at least 13 occasions in our record. (Fig. 7e). These data are also in agreement with the δ¹³C global isotopes oscillations between 26 and 25 Ma (Cramer et al., 2009; Liebrand et al., 2017), that suggest low values for an AABW and high δ¹³C values for a NADW, that may represent the different oceanic primary production and ventilation rates, as proposed in this work. In addition, δ¹³C records in the Atlantic show systematic offsets to lower values toward a North Atlantic signal for most of the late Oligocene to early Miocene. These data suggest the influence of two distinct deep-water sources: cooler southern component water and warmer northern component water (Billups et al., 2002; Pekar et al., 2006; Liebrand et al., 2011). In addition, the increased presence of North Component Deep waters influencing this sector of the eastern Wilkes Land margin could be related with a slowdown of the southern limb of the overturning circulation.

This is also reinforced by several interpretations that document a late Oligocene increase in the influence of North Component Water (e.g. NADW-like) in the Southern Ocean (Billups et al., 2002; Pekar et al., 2006; Villa and Persico, 2006; Scher and Martin, 2008; Liebrand et al., 2011).

These data are also in agreement with the δ¹³C global isotopes oscillations between 26 and 25 Ma (Cramer et al., 2009), that suggest low values for an AABW and high δ¹³C values for a NADW, that may represent the different oceanic primary production and ventilation rates, as proposed in this work. In addition, δ¹³C records on the Atlantic show systematic offsets to lower values toward a North Atlantic signal for most of the late Oligocene to early Miocene. These data suggest the influence of two distinct deep-water sources: cooler southern component water and warmer northern component water (Billups et al., 2002; Pekar et al., 2006; Liebrand et al., 2011). The observed preserved coccolitospheres in the carbonate-rich facies suggest an increased influence of warmer northern component waters to the proto-CDW over the site at least in 13 occasions between 26 and 25 Ma.

**4.4 Orbital forcing and Glacial and Interglacial cyclicity**

The first spectral analysis on late Oligocene sediments from the eastern Wilkes Land margin at Site U1356 shows that glacial-interglacial cycles, resulting in changes in the oceanic configuration off Wilkes Land, are paced with variations in Earth's orbit and seasonal insolation. Although the data is somewhat noisy discontinuous due to gaps in our record, it clearly shows that the glacial-interglacial cyclicity (every 2 m or 41 kyr) discussed above has a persistent obliquity pacing throughout the studied late Oligocene interval (26-25 Ma) in the Wilkes Land. Consequently, this obliquity-paced cyclicity modulates the amount of deep-water production in the Southern Ocean, and exerts a major control on oceanic configuration and current strength. Bottom current velocity fluctuations and ventilation of

bottom sediments respond to the forcings applied by the strength of the Southern Hemisphere westerlies, the position of the PF respect to the site, and consequently by the water mass occupying the bottom of the basin at each time. In addition to obliquity,  precession is also present, which implies a dynamic response of the EAIS and offshore oceanic water masses to orbital forcing.

[revised manuscript text omitted]

Spectral analysis on late Oligocene sediments from the eastern Wilkes Land margin reveal that glacial-interglacial paleoceanographic changes during the late Oligocene are regulated primarily by obliquity, although frequencies in the eccentricity and precession band are also recorded. However, as we do not have a measure of ice dynamics during this time (e.g. ice rafted debris), the orbital response of terrestrial ice in the Wilkes Land Basin remains ambiguous, beyond what is inferred from the deep-sea isotope record.

~~Our record shows that during under the high CO$_2$ values of the late Oligocene (i.e., from ~750 ppm to 400 ppm), ice sheets had retreated to their terrestrial margins, with ice sheet mass loss dominated by surface melt processes. It also indicates a slowdown of the southern limb of overturning circulation,~~

905

**Acknowledgements**

This research used samples and data provided by the Integrated Ocean Drilling Program, now the International Ocean Discovery Program (IODP) . ~~The IODP is sponsored by the US National Science Foundation (NSF) and participating countries under the management of Joint Oceanographic Institutions, Inc. 
[revised manuscript text omitted]

**Supplementary materials**

This file includes:

**S.1 Lithostratigraphy**

**Figures S1, S2**

**S.2 Astrochronologic analysis**

**Figures S3-S10**

**R_analysis**


green claystones with thin silt laminae with planar and cross-bedded laminations presenting different traction and suspension structures (F1). These are interbedded with interglacial highly bioturbated, thicker pale-brown, silty-claytones (F2). This alternation is disrupted from 710 to 730 mbsf by a MTDs  facies (i.e., slupms). From 785 to 879 mbsf  (wthin shipboard lithostratrigraphic unit VIII) slump facies prevail. Slump facies consist predominantly of allochthonous stratified and chaotic sediments of similar lithology to F1 and F2. The interval from 879 to 895 corresponds with lithostratigraphic unit IX described on shipboard. This unit comprises sediments from the middle Eocene and the earliest Oligocene consisting of bioturbated purple silty claystones with some laminations. Erosion/non-deposition surfaces are present within this facies. They are intercalated with coarser green micaceous (very shiny) (sandy) silty-claystone. Laminations with ripples and pinstripe and cross-lamination are also observed. This facies are intercalated with MTD facies composed of these same sediments. The interval between 895 and 896 mbsf is within shipboard lithostratigraphic unit X and is characterized by a lithological change to Eocene green sands Facies.

[Figure]

[Figure]

Site U1356
IODP Exp. 318
Cores 42-95R

**LEGEND U1356 CORES 42R-95R**

Brown silty-claystone

Green claystone with thin silt laminae

Silt laminae, with cross and/or parallel laminations

Brown, reddish claystones

Silty matrix with claystone clasts. Carbonated cement.

Very dark grey claystone

Dark grey claystone

Brown claystone with silica diagenesis. Black belts. Barren. Banded.

Dark golden greenish silty clay. Drilling disturbed.

White clay. Barren.

Black claystone.

Silt laminations with cross bedding, starved ripples, planar laminations.

Clay bearing nannofossil ooze. Carbonate cemented.

Claystone to siltstone with pyrite.

Dark green claystone, no bioturbation, pyrite.

Massive nanno rich dark claystone

Silty claystone, carbonate cemented.

Light brown green silty claystone

Gravel

Dark green clay very deformed and fractured, no clasts.

Red claystone, deformed, no clasts.

PM: Paleomagnetic age
CN: Calcareous Nannofossils
F: Foraminifers
PL: Dinocysts

Matrix supported silty sand with mud clasts and pebbles. Deformed and chaotic.

Matrix supported silty sand with mud clasts and pebbles. Deformed and chaotic.

Matrix supported of sandy silt, rich in dropstones. Convoluted sediment. Mud clasts. Sand dikes.

Matrix supported silt with clayey mud clasts. Contorted. Dropstones in coarse material.

Silty clay matrix supported with sand dikes. Dropstones and mud clasts present.

Silt clay matrix supported with mud clasts and few dropstones.

Inclined laminations, deformed, with dropstones.

Clasts of silty debris flow with dropstones.

Clast.

Breccia/conglomerate with mud clasts.

Reddish brown sandy matrix supported which high density of clasts up to 3cm Ø. Mud clasts. Traction struct.

Grey silty matrix, sort poor, heterogeneous sizes, ~0.1-2cm Ø.

Sandy silt matrix. Clast rich, homogeneous sizes, ~2mm Ø.

Purple silty claystone.

Green micaceous (very shiny) (sandy) silty-claystone. Laminations with ripples and pinstripe and cross lamination.

Slump mixing of the green silty(sandy) claystone with the purple silty claystone. IRD, rock fragments.

core width =
coarser deposits

Microfaulting

"Pseudonodules" - load structures that have become disconnected from their source bed.

Shell lag

Bioturbation

soft sediment mixing, folding

Sand dikes

Cross and parallel lamination

T : Turbidites: levee beds with Bouma b,c,d or c,d sequences

C : Contourites: isolated flat & cross laminations.

**FACIES**

F1 — Laminated Facies "Glacial?"

F2 — Bioturbated Facies "Interglacial?"

Slumps Facies

Debris Flows Facies

Carbonate cemented beds

Turbidite type Facies

Miocene Facies Turbidites and hemipelagites

EOT Facies Slumps – debris flows

EOT Facies Bioturbated siltstones and laminated claystones

Eocene Sands Facies

**Fig. S1**: Detailed sedimentary log from IODP U1356 Site U1356 exp. 318 from 11R to 95R (95.4 to 896 mbsf).

[Figure]

[Figure]

**U1356**
**IODP Exp. 318**
**Cores 11R-95R**
**Facies Log**

**FACIES**

**Miocene**

- Laminated Green Silty clay diatom ooze "Glacial"
- Green silty clay diatom ooze intercalated with intervals of grey silty clays diatom ooze
- Grey silty clays diatom ooze "Interglacial"
- **F2** Bioturbated Facies "Interglacial"
- Debris flows with low to abundant clasts
- Turbidite type Facies

**Oligocene**

- **F1** Laminated Facies "Glacial"
- **F2** Bioturbated Facies "Interglacial"
- Slumps Facies
- Debris Flows Facies
- Carbonate cemented beds

**Eocene**

- EOT Facies Slumps - debris flows
- EOT Facies Bioturbated siltstones and laminated claystones
- Eocene Sands Facies

▶ Paleomagnetic Tie point Age (Ma) and Chron (Tauxe et al., 2012) GPTS2012

**Fig. S2**: IODP Site U1356 Exp. 318 from 11R to 95R (95.4 to 896 mbsf). Schematic Facies Log with plotted Magnetic Susceptibility (MS). Paleomagnetic tie points are also present. Ages from Tauxe et al., (2012) are updated to GPTS 2012.

**S.2 Astrochronologic analysis**

**Materials and methods**

We followed the procedures published by Meyers et al. (2012) and Wanlu Fu et al. (2016) in order to generate spectral analysis on our data.

We selected Zr/Ba ratio as we consider this ratio to integrate and summarize the processes shaping our facies model, showing clearly the marked cyclicity present.

Data preparation:

In order to remove the long-term trend data series were detrended, outliers were removed, and sampling interval was linearly interpolated in order to resample the dataset to an even spacing of 2cm. Average sedimentation rates between the two paleomagnetic tie end points were linearly interpolated and is 5cm/kyr for the investigated interval. Age model is calibrated to the Geologic Timescale 2012 (GPTS 2012, Table 1).

For initial cyclostratigraphic analyses, we used Anlyseries software (Paillard et al., 1996). We used B-Tukey method in order to preliminary assess the cyclicity on the record on a depth scale. A clear and statistically significant cyclicity is observed in Ba, Zr and Zr/Ti every 2m (0.5 cycles/m), and less significant ones but also reliable at 4.67m (0.21 cycles/m), and 1m (0.94 cycles/m) (Fig. S3). On the basis of the calculated sedimentation rate, the cycles above (0.5 cycles/m) account for 40 Kyr. After determining the significant frequency, we filtered the Zr/Ba dataset (at depth domain) at 0.5 frequency in order to extract the wavelet and compare it with the obliquity solution for that time-period (Laskar et al., 2004). Cycles can be correlated one to one with a total of 23 cycles of obliquity (Fig. S3). After initial analysis we proceed with Astrochron Evolutive Average Spectral Misfit method (Meyers et al., 2012). Astrochron package is prepared to resolve unevenly sampled series, and changing sedimentation rates.

Time-frequency analysis:

Evolutive Harmonic Analysis (EHA; Fig. S4) of the prepared Zr/Ba (in depth scale) data provides an evaluation of changes in the spectral features through depth/time. EHA employs five $3\pi$ DPSS tapers, and a moving window of 15 m. Significant frequencies are retrieved for further study.

Astrochronologic testing:

The Evolutive Average Spectral Misfit method (Meyers, 2014) (E-ASM; five $3\pi$ tapers; searching to the mean Nyquist frequency of 1.504221 cycles/m) was used to test a range of plausible timescales and simultaneously evaluate the reliability of the presence of astronomical cycles. The ETP (eccentricity, obliquity and precession) target periods were determined from La04 (Laskar et al., 2004) using the interval from 25.0 – 26.4 Ma: 400.00 kyr (E1), 131.58 kyr (E2), 99.01 kyr (E3), 40.49 kyr (O1), 32.79 kyr (O2), 20.70 kyr (P1), 19.69 kyr (P2) and 17.06 kyr (P3).

The Zr/Ba MTM Harmonic F-test results of the EHA (Fig. 6) are evaluated using a grid of 100 sedimentation rates spanning 3 cm/Kyr to 10 cm/Kyr. This range of sedimentation rates encompasses the long-term average sedimentation rate for the section based on available paleomagnetic constraints with a total duration of 0.71 Ma and a total stratigraphic thickness between 30 – 40 m given the range of plausible correlation horizons for the section. We interpolate an average sedimentation rate of 5 cm/Kyr.

All spectral peaks above 90% F-test confidence level were evaluated using E-ASM, and Monte Carlo significance testing utilizing 10,000 simulations. Results with Null Hypothesis Significance Levels (Ho-SL) less than or equal to 0.1% were identified (Fig. S5).

Astronomical tuning:

Frequency domain minimal tuning (Meyers et al., 2001) is used for tracking obliquity in EHA harmonic F-test for the calibrated periods and sedimentation rates. Spatial frequencies are afterwards converted to sedimentation rates using average period of 41 Kyr (Fig. S6) and a time-space map is created with a new calibrated time series. Time series is afterwards anchored to our paleomagnetic tie points (Fig. S7).

MTM and EHA results on the tuned data provide further evidence of the presence of precession, obliquity and eccentricity cycles, supporting the obliquity tuning (Fig. S8).

Significant Harmonic F-test peaks that achieve 95% CL are: 102.77 Kyr, 69.26 Kyr, 40.84 Kyr, 30.05 Kyr, 22.27 Kyr, 20.68 Kyr (Fig. S8).

The new time-scale is used to tune the other records (Zr/Ti; Ca/Ti; MS; Ba). EHA is then applied to tuned records in order to see the frequencies that appear (Fig. S9).

Orbital frequencies were tested in each core section individually in the Zr/Ba dataset in the depth scale in order to assure that cyclicity is not an artefact related to the gaps in the series (Fig. S10).

[Figure]

**Fig. S3:** Tuning of Zr/Ba to the obliquity solution. Zr/Ba (in the depth domain) is tuned to the obliquity solution cycle by cycle. Tuning of Zr/Ba record (in depth scale) and bandpass filtering were done in Analyseries (Paillard et al., 1996). (a) Magnetostratigraphic chrons (Tauxe et al., 2012); (b) schematic stratigraphic log; (c) Zr/Ba data in depth scale with the envelope filter centred at 2m (in red); (d) Zr/Ba data in time scale using paleomagnetic tie points and a linear sedimentation rate, with the envelope filter centred at 40 Kyr (in red); (e) Eccentricity and obliquity solutions (Laskar et al., 2004); (f) Polarity

chrons from the GPTS2012; (g) Blakmann-Tukey in the Zr/Ba data in time domain (not tuned), with statistical (>90%) periodic peaks in the 40 Kyr and 20 Kyr periodicities; g) Blakmann-Tukey in the Zr/Ba data in depth domain, with statistical (>90%) periodic peaks.

[Figure]

**Fig. S4:** Evolutive Harmonic Analysis (EHA) in depth scale Zr/Ba data. The detrended Zr/Ba data is linearly interpolated to a constant sample spacing of 2 cm prior to analysis. EHA employs with 3DSDP tapers and a 15 window.

[Figure]

**Fig. S5:** Astrochronologic testing using evolutive ASM analysis. (a) Evolutive ASM plot, displaying Ho-SL values (90% confidence level), across sedimentation rates spanning 3 to 10 cm/kyr. (b-c) Summary of evolutive ASM results, using a threshold Ho-SL value of 0.1 to identify optimal sedimentation rates. (b) displays each sedimentation rate, and (c) displays the associated Ho-SL.

[Figure]

**Fig. S6:** Frequency tracking for minimal tuning. The obliquity cyclicity (41 Kyr) can be tracked in the EHA harmonic F-test confidence level results by setting the fmin=0.01 and fmax=0.4 based on the spatial frequencies calculated by EASM results. Calculated sedimentation rates based on spatial frequency tracking.

[Figure]

**Fig. S7:** Tuned record and depth-time plot derived by frequency domain minimal tuning to the obliquity cycle. The red dotted lines show the correlated depth and time.

[Figure]

**Fig. S8:** MTM results of the tuned data with the major periods in kyr. These peaks achieve the 95% confidence level for both the MTM harmonic F-test and the AR1 red noise model or the AR1 noise model only.

[Figure]

**Fig. S9:** Following Zr/Ba tuning (a), MS (b), Zr/Ti (c), Ba (d), and Ca/Ti (e) have been tuned. EHA analysis was applied in order to depict the frequencies. EHA on the ETP solution is also added in order to compare the resulting frequencies (f).

[Figure]

**Fig. S10:** Spectral analysis over individual core sections for the Zr/Ba dataset on the depth scale. The 0.5 cycle/m (that counts for obliquity) achieve >90% significance in all cores except for core 70 where smaller frequencies seem to dominate. Changes in peak frequency seem to be dominated by slightly changes in sedimentation rate. Each core was analysed by a) MTM and b) EHA with a 5 m window.

**R_analysis**

```
######################################
**(1) LOAD THE R-PACKAGE 'ASTROCHRON'**
######################################
library(astrochron)

######################################
**(2) READ DATA FILE**
######################################
**Read the carbon isotope data from file 'CarbonIso.csv'**
dat<-read(d=0)

#########################
**(3) PREPARE TIME SERIES**
#########################
**The median sampling interval of the prepared data is 0.02 m, and the mean**
sampling interval is 0.023 m
**Resample Zr/Ba data to 2.5 cm sampling grid, using piecewise linear**
interpolation
ZrBa<- linterp(dat,dt=0.025)

#########################################
**(4) PERFORM EVOLUTIVE HARMONIC ANALYSIS**
#########################################
mtmML96(ZrBa,xmax=2,pl=2,siglevel=.90,sigID=T)
mtm(ZrBa,xmax=2,pl=2,sigID=T)
**Use a 12 meter window, with five 3pi DPSS tapers.**
**Search up to the mean Nyquist frequency of 1.504221 cycle/m**
**Output F-test confidence level estimates for evolutive average spectral misfit (ASM)**
analysis.
prob=eha(ZrBa,fmax=2,output=4,genplot=4,pl=2,ydir=-1,win=15)

###############################################################
**(5) IDENTIFY TARGET PERIODS FOR AVERAGE SPECTRAL MISFIT ANALYSIS**
###############################################################
**Obliquity and precession terms from Laskar et al. (2004)**
model=etp(tmin=25000,tmax=26400)
eha(model,win=200,fmax=0.1,sigID=T,pad=10000)
mtm(model,xmax=0.1,pl=2,sigID=T)

#################################################
**(6) EVOLUTIVE AVERAGE SPECTRAL MISFIT ANALYSIS**
#################################################
**Set up analysis parameters:**
**Astronomical target frequencies are determined from Laskar et al. (2004)**
target=c(1/404,1/124,1/95,1/54,1/41,1/29,1/23,1/19)
**Ray=(1/N*Ax) on N= number of points in data series and Ax= sampling resolution of data**
series
**Use average sampling interval to estimate the Nyquist frequency (for 1/(0.025m sampling**
2) Fnyq=(1/2*Ax)
rayleigh=0.0217
nyquist=20
**Average sedimentation rates is around 5cm/Kyr**
**The total duration between the youngest and oldest age is 0.71 Ma +/- ? Ma.**
```

```
**Given the range of plauisble correlation horizons into the LO section,**
**the total stratigraphic thickness can range from 30 m to 40 m.**
**Execute evolutive ASM analysis. This will take 10-20 minutes to complete.**
res1=eAsm(prob,target=target,rayleigh=rayleigh,nyquist=nyquist,sedmin=3,sedmax=10,numsed
=100,siglevel=0.95,iter=10000,output=4)
**Track Ho-SL minima from evolutive ASM results**
**Identify those results with Ho-SL less than or equal to 0.1%**
pl(1); eAsmTrack(res1[1],threshold=0.1,ydir=1)

############################################################
**(7) EXAMINE SELECTED SPECTRA AND ASM-CALIBRATED PERIODS**
############################################################
**F-test CL spectrum from 43.965 m**
**Calculate calibrated periods in kyr (observed)**

prob_674.53=extract(prob,get=674.53)
1/(peak(prob_674.53,level=0.9)[2]*0.025)

prob_670.03=extract(prob,get=670.03)
1/(peak(prob_670.03,level=0.9)[2]*0.025)

prob_656.03=extract(prob,get=656.03)
1/(peak(prob_656.03,level=0.9)[2]*0.025)

prob_661.03=extract(prob,get=661.03)
1/(peak(prob_661.03,level=0.9)[2]*0.025)

#############################################################
**(8) ASTRONOMICALLY-TUNE Zr/Ba DATA USING**
**FREQUENCY-DOMAIN MINIMAL TUNING (Meyers et al., 2001)**
#############################################################
**Track obliquity in EHA harmonic F-test confidence level given the ASM-calibrated periods**
**Track obl term on the basis of the ASM calibrated sedmentation rates**
**Note that the Rayleigh frequency is 0.0217 cycles/m**
freqs=trackFreq(prob,fmin=0.023,fmax=1,threshold=0.9)
**convert spatial frequencies to sedimentation rates using average period of 41 kyr**
sedrate=freq2sedrate(freqs,period=41,ydir=-1)
sedrate
**View the calibrated sedimentation rates on depth sedrate**
**Integrate the sedimentation rate curve to create a time-space map**
time=sedrate2time(sedrate)
**View the calibrated time series**
time

**The duration of specific interval can be calculated by the output of sedrate and time**
**Tune the ZRBA series using the time-space map**
tuned=tune(ZrBa,time)
#############################################################
**(9) PREPARE TUNED SERIES AND EVALUATE SPECTRA**
#############################################################
**Interpolate the tuned series. Median sampling interval is 0.5 kyr and mean is 0.55 kyr.**
**Will use AR1 test; use a conservative interpolation to avoid introducing serial correlation.**
datatuned=linterp(tuned,dt=0.6)
**Perform MTM analysis on the tuned series**
spec=mtm(datatuned,tbw=2,pl=2,siglevel=0.95,xmax=0.06,output=1,sigID=T)
```

```
**identify periods of AR1 CL peaks that acheive the 90% AR1 CL**
1/peak(cb(spec,c(1,4)),level=95)[2]
**Perform EHA on the tuned series**
ZrBa_final<-read(d=0)
pwr=eha(datatuned,fmax=0.06,output=2,ydir=-1,win=250)
plotEha(pwr,pl=1,ydir=-1)

############################################################
**(10) BANDPASS FILTERING AND ECCENTRICITY AMPLITUDE**
MODULATION ANALYSIS
############################################################
**Perform bandpass-filtering on the tuned series to extract short eccentricity (E2+E3)**
e23_data=bandpass(datatuned,flow=0.006,fhigh=0.011,xmax=0.02)
**Perform bandpass-filtering on the eccentricity terms from Laskar et al. (2011)**
model=getLaskar("la10d")
model=iso(model,xmin=25000,xmax=26400)
e23_model=bandpass(model,flow=0.006,fhigh=0.011,xmax=0.02) #short-term
eccentricity
**Evaluate the alignment between the amplitude envelope of the filtered short-term eccentricity**
and the filtered long-term eccentricity
am_data=hilbert(bandpass(datatuned,flow=0.006,fhigh=0.011))
pl(1)
plot(s(am_data),type="l",ylim=c(-3,3))
lines(s(e1_data),col="red")

**Anchor the tuned data to a tie point from the time scale where 606.226 is equivalent to 678.78**
ZrBa_tuned2=anchorTime(datatuned,606.2262,25990,flipOut=T,timeDir=2)

write.csv(ZrBa_tuned2,file="ZrBa_tuned_at_606.226to25990.csv") #Save data

Tune other data series and analyse with EHA:

Ba<-read(d=0)
Ba_tuned=tune(Ba,time)
eha(linterp(Ba_tuned,dt=0.5),fmax=0.06,output=2,ydir=-1,win=150)
Ba_tuned=anchorTime(datatuned,678.9,25990,flipOut=T,timeDir=2)

MS<-read(d=0)
MS_tuned=tune(MS,time)
eha(linterp(MS_tuned,dt=0.5),fmax=0.06,output=2,ydir=-1,win=150)
MS_tuned=anchorTime(datatuned,678.9,25990,flipOut=T,timeDir=2)
```